# Integrative proteogenomic characterization of hepatocellular carcinoma across etiologies and stages

Charlotte K. Y. Ng [1,2,3,4], Eva Dazert [5,9], Tuyana Boldanova[1,6,9], Mairene Coto-Llerena [1,2], Sandro Nuciforo [1], Caner Ercan [2], Aleksei Suslov[1], Marie-Anne Meier[1], Thomas Bock [5], Alexander Schmidt [5], Sylvia Ketterer[1], Xueya Wang[1], Stefan Wieland [1], Matthias S. Matter [2], Marco Colombi[5], Salvatore Piscuoglio [1,2], Luigi M. Terracciano[2,7,8,10], Michael N. Hall[5,10] & Markus H. Heim [1,6,10 ✉]

Proteogenomic analyses of hepatocellular carcinomas (HCC) have focused on early-stage, HBV-associated HCCs. Here we present an integrated proteogenomic analysis of HCCs across clinical stages and etiologies. Pathways related to cell cycle, transcriptional and translational control, signaling transduction, and metabolism are dysregulated and differentially regulated on the genomic, transcriptomic, proteomic and phosphoproteomic levels. We describe candidate copy number-driven driver genes involved in epithelial-to-mesenchymal transition, the Wnt-β-catenin, AKT/mTOR and Notch pathways, cell cycle and DNA damage regulation. The targetable aurora kinase A and CDKs are upregulated. *CTNNB1* and *TP53* mutations are associated with altered protein phosphorylation related to actin filament organization and lipid metabolism, respectively. Integrative proteogenomic clusters show that HCC constitutes heterogeneous subgroups with distinct regulation of biological processes, metabolic reprogramming and kinase activation. Our study provides a comprehensive overview of the proteomic and phophoproteomic landscapes of HCCs, revealing the major pathways altered in the (phospho)proteome.

[1] Department of Biomedicine, University Hospital Basel, University of Basel, Basel, Switzerland. [2] Institute of Medical Genetics and Pathology, University Hospital Basel, University of Basel, Basel, Switzerland. [3] Department for BioMedical Research (DBMR), University of Bern, Bern, Switzerland. [4] SIB Swiss Institute of Bioinformatics, Lausanne, Switzerland. [5] Biozentrum, University of Basel, Basel, Switzerland. [6] Department of Gastroenterology and Hepatology, University Hospital Basel, Basel, Switzerland. [7] Department of Pathology, Humanitas Clinical and Research Center, IRCCS, Milan, Italy. [8] Department of Biomedical Sciences, Humanitas University, Milan, Italy. [9] These authors contributed equally: Eva Dazert, Tuyana Boldanova. [10] These authors jointly supervised this work: Luigi M. Terracciano, Michael N. Hall, Markus H. Heim. ✉email: markus.heim@unibas.ch

Liver cancer, of which 75–85% are hepatocellular carcinoma (HCC), caused 782,000 deaths globally in 2018[1]. Genomic analyses have revealed that *TERT* promoter, *CTNNB1* (encoding β-catenin) and *TP53* (encoding p53) are frequently mutated in HCC, while genes involved in other critical processes, such as oxidative stress response, chromatin remodeling and hepatocyte differentiation, are recurrently mutated but in <10% of HCC[2–4]. Transcriptomic subtyping has revealed HCC subclasses that differ in the expression of genes related to proliferation, stemness, metabolism, hepatocyte differentiation and liver function[5–10]. More recently, three mass spectrometry-based proteogenomic studies of hepatitis B virus (HBV)-associated HCCs have been published[11–13]. In the first study, the proteome profiling of early-stage HBV-associated HCCs found that a subset of HCCs characterized by disrupted cholesterol homeostasis and SOAT1 overexpression was associated with poor outcome[11]. Indeed, avasimibe, a SOAT1 inhibitor, effectively reduced the size of tumors overexpressing SOAT1 in patient-derived xenograft models[11]. In the second study, integrated proteogenomic analysis of HBV-related HCC[12] revealed three proteome subclasses, namely metabolism, proliferation and microenvironment dysregulated subgroups, that were associated with patient survival, tumor thrombus and genetic profile. PYCR2 and ADH1A, both implicated in metabolic reprogramming in HCC, were further identified as proteomic prognostic biomarkers. In the third study, the authors found significant intratumor heterogeneity on the genome and transcriptome levels but patient specificity on the proteome and metabolome levels among resected and predominantly early-stage and HBV-associated HCCs[13]. While these studies have provided great insights into the proteome of HCCs that are primarily early-stage and HBV-associated, relatively little of their phosphoproteome was described. In this work, we present an integrated (phospho)proteogenomic analysis of HCC biopsies across etiologies and clinical stages, representative of the wide spectrum of molecular heterogeneity of HCC. We show that HCC are underpinned by the dysregulation of oncogenic processes related to transcriptional and translational control, metabolism, the Wnt-β-catenin, AKT/mTOR and Notch pathways, which are differentially regulated on the genomic, transcriptomic, proteomic and phosphoproteomic levels.

## Results

**Proteogenomic profiling of HCC.** We collected biopsies from 122 tumors from 114 patients, including seven patients with >1 multicentric (genetically independent) tumors (Table 1, Supplementary Data 1 and Supplementary Fig. 1). 94% of the patients had at least one underlying liver disease, primarily alcohol liver disease (59%) and/or hepatitis C infection (26%). 53% of the patients were early stage (BCLC 0/A) and 47% were more advanced stage (BCLC B/C/D). None of the patients had undergone systemic therapy for their disease. We performed whole-exome sequencing and RNA-sequencing for all 122 tumors and global proteome and phosphoproteome profiling using liquid chromatography-tandem MS analyses on a subset of 51 tumors (Table 1, Supplementary Figs. 2 and 3 and Supplementary Data 1). The proteome was measured in data-independent manner by selected window acquisition of theoretical mass (SWATH)[14,15] and the phosphoproteome was measured in data-dependent and label-free manner. The subset of 51 tumors subjected to proteome and phosphoproteome profiling was representative of the entire cohort in terms of clinicopathological parameters and molecular profiles (Table 1 and Supplementary Fig. 4). As controls, we also performed RNA-sequencing, proteome and phosphoproteome profiling on 15, 11 and 10 normal biopsies from individuals without HCC and with normal liver values, respectively (Supplementary Data 1).

**Deregulated pathways in HCC.** Principal component analyses showed that HCCs are distinct from and more variable than normal livers on the transcriptome, proteome and phosphoproteome levels (Fig. 1a–c). HCCs did not segregate according to their underlying liver diseases (Supplementary Fig. 5). Across all molecular levels, low-grade HCCs were more homogeneous than high-grade HCCs as measured by intra-group variability (Spearman rho between 0.26 and 0.37, all $p < 0.0001$, Fig. 1d). In accordance with the definition of histological (Edmondson) grading, we also found that low-grade HCCs were more similar to normal livers than high-grade HCCs (Spearman rho between 0.51 and 0.66, all $p < 0.0001$, Fig. 1e). These observations were corroborated by an analysis of The Cancer Genome Atlas data (Supplementary Fig. 6).

To identify pathways deregulated in HCC, we performed differential expression analyses of the HCC transcriptome and proteome compared to normal livers. We observed a moderate correlation between the deregulation of the transcriptome and the proteome (Spearman rho = 0.33, $p < 0.0001$, Fig. 1f). A quadrant analysis of transcriptome and proteome data showed that 37.7% (15.7% for adjusted $p \leq 0.05$) of genes were up-regulated on both the mRNA and the protein levels, 20.9% (6.2%) were up-regulated on the mRNA level but down-regulated on the protein level, 16.6% (5.4%) were upregulated in protein but down-regulated on the mRNA level, and 24.8% (13.6%) were downregulated in both. Pathway analysis of these four quadrants revealed that the genes/proteins up-regulated on both the mRNA and the protein levels were enriched in pathways related to mRNA splicing, epigenetic regulation of rRNA expression and translation (Fig. 1f and Supplementary Data 2). By contrast, genes/proteins consistently down-regulated were enriched in metabolism pathways of amino acids, fatty acids, and other metabolites. Among genes upregulated only on the mRNA level, pathways related to translational control, proteasome and oxidative phosphorylation were enriched. By contrast, pathways related to the complement and coagulation were enriched among proteins upregulated only on the protein level.

Previous proteogenomic studies focused on surgically resected HBV-associated or early-stage HCCs[2,11,12], we therefore asked whether early and late-stage HCC would show different molecular characteristics. We observed no difference in terms of the frequency of mutated genes (Supplementary Fig. 7). On the transcriptomic and proteomic levels, high-stage HCCs overexpress genes and proteins related to cell cycle and mitosis, DNA repair and replication, transcriptional regulation (Supplementary Data 3). On the other hand, high-stage HCCs underexpress genes related to ECM formation and organization on the mRNA level and metabolism of fatty acids on the protein level.

Taken together, while we observed overall upregulation of pathways related to mRNA splicing and downregulation of pathways related to normal liver function, we also observed translational control-related pathways being upregulated on the mRNA level only, and pathways related to coagulation and the complement upregulated on the protein level only.

**CNA-mRNA-protein correlation analysis identifies candidate driver genes.** Next, we evaluated the correlation between copy number alteration (CNA), mRNA expression and protein expression. The median CNA-mRNA and mRNA-protein Spearman correlation coefficients were 0.203 and 0.287, respectively (Fig. 2a, b). We similarly observed a higher fraction of genes with significantly positive (Spearman rho >0.3) mRNA-protein correlation than for CNA-mRNA correlation (45.1% vs 32.0%), with the latter higher than previously reported[13]. Gene set enrichment analysis of CNA-mRNA and mRNA-protein

**Table 1 Summary of clinicopathological information of the cohort.**

| | | Cohort with genomic and transcriptomic data (122 biopsies from 114 patients) | | Cohort with genomic, transcriptomic, proteomic and phosphoproteomic data (51 biopsies from 49 patients) | | Comparison between biopsies with (n = 51) and without (n = 71) complete molecular profiling |
|---|---|---|---|---|---|---|
| | | n | (%) | n | (%) | |
| Sex (n = 114, 49) | Male | 97 | 85% | 41 | 84% | ns |
| | Female | 17 | 15% | 8 | 16% | |
| Age at classification (n = 114, 49) | (median, range) | 69 (18–87) | | 66 (18–84) | | ns |
| BCLC (n = 115, 49)[a] | 0 | 4 | 3% | 1 | 2% | ns |
| | A | 57 | 50% | 25 | 51% | |
| | B | 27 | 23% | 14 | 29% | |
| | C | 24 | 21% | 7 | 14% | |
| | D | 3 | 3% | 2 | 4% | |
| Number of tumors (n = 115, 49)[a] | 1 | 53 | 46% | 24 | 49% | ns |
| | 2 | 20 | 17% | 7 | 14% | |
| | 3 | 6 | 5% | 3 | 6% | |
| | 4 | 1 | 1% | 0 | 0% | |
| | 5 | 1 | 1% | 0 | 0% | |
| | multinodular | 34 | 30% | 15 | 31% | |
| Macrovascular invasion (n = 115, 49)[a] | yes | 17 | 15% | 5 | 10% | ns |
| | no | 98 | 85% | 44 | 90% | |
| Metastasis (n = 115, 49)[a] | yes | 11 | 10% | 3 | 6% | ns |
| | no | 104 | 90% | 46 | 94% | |
| Child-Pugh (n = 115, 49)[a] | A | 69 | 60% | 26 | 53% | ns |
| | B | 40 | 35% | 18 | 37% | |
| | C | 3 | 3% | 2 | 4% | |
| | (na/nd) | 3 | 3% | 3 | 6% | |
| MELD (n = 115, 49)[a] | (median, range) | 9 (5–25) | | 9 (6–24) | | ns |
| | (na/nd) | 2 | 2% | 2 | 4% | |
| Cirrhosis (n = 115, 49)[a] | yes | 83 | 72% | 35 | 71% | ns |
| | no | 32 | 28% | 14 | 29% | |
| Underlying liver disease (n = 115, 49)[a,b] | Hepatitis B | 13 | 11% | 7 | 14% | ns |
| | Hepatitis C | 30 | 26% | 18 | 37% | p = 0.04 |
| | Alcoholic liver disease | 68 | 59% | 25 | 51% | ns |
| | Non-alcoholic fatty liver disease | 19 | 17% | 9 | 18% | ns |
| | No liver disease | 7 | 6% | 2 | 4% | ns |
| Edmondson grade (n = 122, 51) | 1 | 7 | 6% | 5 | 10% | ns |
| | 2 | 66 | 54% | 25 | 49% | |
| | 3 | 41 | 34% | 16 | 31% | |
| | 4 | 8 | 7% | 5 | 10% | |
| Immunophenotype (n = 122, 51) | Inflamed | 37 | 30% | 13 | 25% | ns |
| | Immune-excluded | 43 | 35% | 21 | 41% | |
| | Immune-desert | 38 | 31% | 14 | 27% | |
| | (na/nd) | 4 | 3% | 3 | 6% | |

Statistical comparisons were performed using Fisher's exact tests (for categorical data with two levels), Chi-squared tests (for categorical data with >2 levels), and two-sided Mann–Whitney U tests (for numerical and ordinal data).
BCLC Barcelona Clinic Liver Cancer clinical staging system, MELD model for end-stage liver disease, na not available, nd not determined; ns not significant.
[a]Determined at the time of biopsy. One patient was biopsied twice seven years apart.
[b]Patient may have >1 underlying liver disease.

correlation revealed 275 and 45 Reactome pathways, with only one (protein localization) enriched in both analyses. The pathways enriched among genes with high CNA-mRNA correlation include RNA transport, ubiquitination and proteasome degradation, transcriptional regulation by *TP53*, translation, cell cycle and DNA repair, and cellular response to stress (Fig. 2c and Supplementary Data 4). By contrast, genes with high mRNA-protein correlation are enriched in pathways related to the metabolism of amino acids, glucose, fatty acids, and xenobiotics (Fig. 2c and Supplementary Data 4).

Genome-wide copy number analysis by GISTIC2 identified 8 recurrently amplified peaks and 11 recurrently deleted peaks (Supplementary Fig. 8 and Supplementary Data 4), 5 of which were enriched among genes with high CNA-mRNA correlation

but none was enriched among genes with high mRNA-protein correlation (Fig. 2d and Supplementary Data 5). One could hypothesize that HCC driver genes would be overrepresented among *cis*-copy number-regulated genes (i.e. the CNA impacts its own expression) that also show high mRNA-protein correlation. To identify such candidate driver genes, we focused on the 136 genes that showed high CNA-mRNA and mRNA-protein correlation (Spearman rho>0.5) and specifically on the 29 within the 5 enriched GISTIC2 regions, which were gained or lost in 27–61% of the cohort (Fig. 2d inset and Supplementary Data 4 and 5). We further narrowed down this list of 29 genes to those that were dysregulated in the expected orientation (i.e. upregulated in amplified regions and downregulated in deleted regions with respect to normal tissues, FDR < 0.05). Of these 29, 19 were

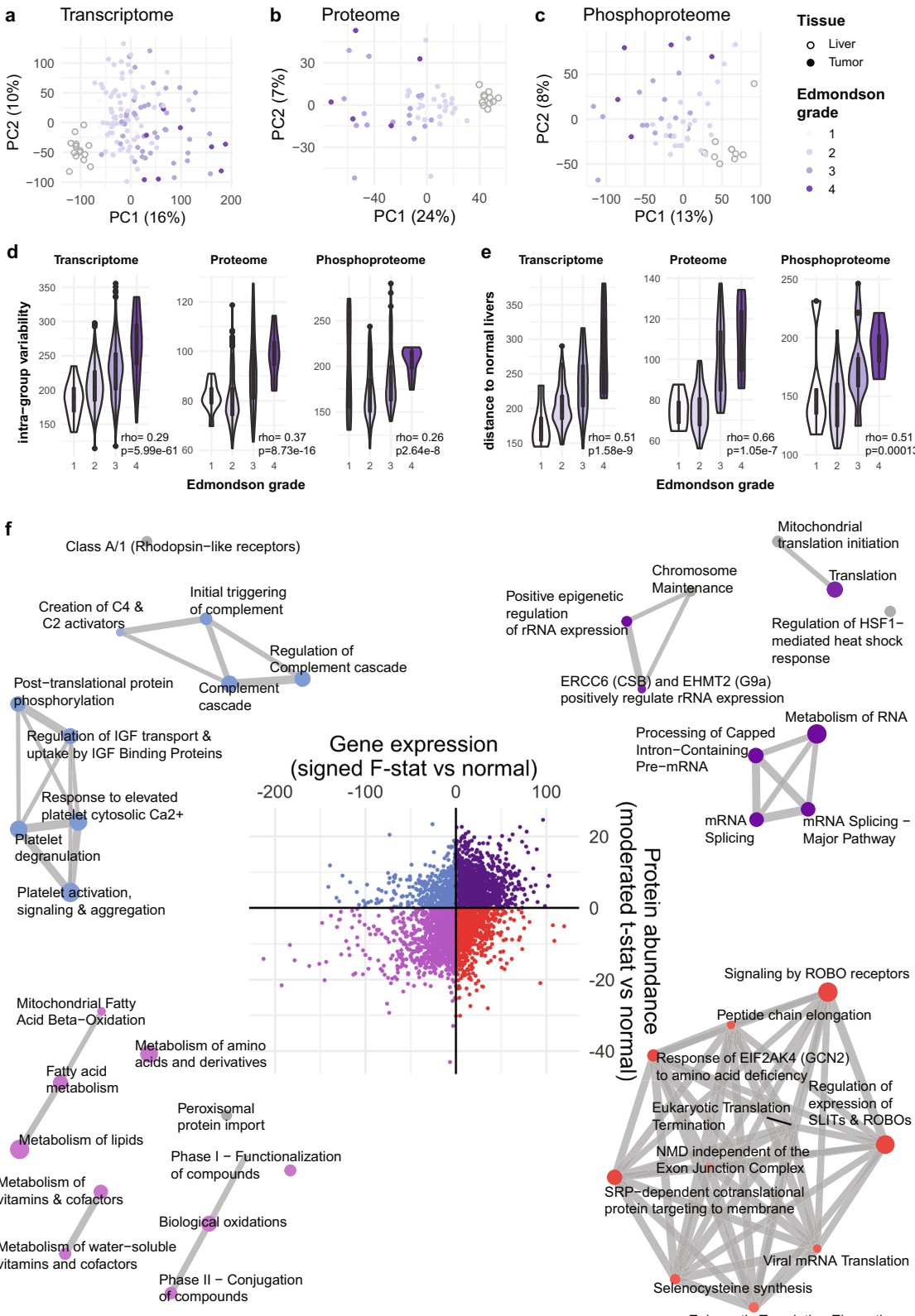

dysregulated on the mRNA level, 11 were dysregulated on the protein level, and 9 were dysregulated on both levels (*ATP6V1C1*, *BYSL*, *CHD1L*, *NUDCD1*, *LRRC47*, *RRM2B*, *UBQLN4*, *XPO5*, *YWHAZ*). Among this group of genes were known cancer genes such as *CHD1L* (Chromodomain Helicase DNA Binding Protein 1 Like, 1q21.3 peak)[16] and *YWHAZ* (14-3-3 zeta, 8q22.2)[17,18] (Fig. 2d inset).

There were other candidate copy number-driven cancer genes implicated in oncogenesis (Fig. 2d inset and Supplementary Data 5). For example, *NUDCD1* (NudC domain containing 1, or OVA66, 8q22.2) has been shown to promote colorectal carcinogenesis and metastasis by inducing epithelial-to-mesenchymal transition (EMT) and inhibiting apoptosis[19] and to promote oncogenic transformation by hyperactivating the

**Fig. 1 Deregulated pathways in HCC. a–c** Principal component analysis plots of (**a**) transcriptome (grade 1 $n = 7$, grade 2 $n = 66$, grade 3 $n = 41$, grade 4 $n = 8$), **b** proteome (grade 1 $n = 5$, grade 2 $n = 25$, grade 3 $n = 16$, grade 4 $n = 5$), **c** phosphoproteome (grade 1 $n = 5$, grade 2 $n = 25$, grade 3 $n = 16$, grade 4 $n = 5$) of HCC biopsies (colored by Edmondson grade) and normal liver biopsies. **d** Intra-group (within Edmondson grade) variability as measured by pairwise Euclidean distance between samples according to principal components (sample size as in (**a–c**)). **e** Distance of each HCC to the median of normal livers as measured by Euclidean distance according to principal components. **d**, **e** Statistical comparisons were performed using Spearman's correlation tests. Thick middle line in the boxplot denotes the median; box extends to the 1st and 3rd quartiles; whiskers extend to the ±1.5 IQR of the box; dots depict the outliers. **f** Scatter plot of (y-axis) the moderated t-statistics from the differential protein expression analysis of HCC vs normal liver against (x-axis) the F-statistics from the differential gene expression analysis of HCC vs normal liver. Points are colored according to the four quadrants. Enrichment maps show the top 10 enriched Reactome pathways from over-representation tests of the genes/proteins in each of the four quadrants. In each enrichment map, gene sets with overlapping gene sets are joined by edges. Nodes are colored according to p-value, where gray indicates a higher p value and dark blue/violet/purple/red indicates a lower p value. The size of the nodes is proportional to the number of genes in the quadrant within a given gene set. Source data are provided as a Source Data file.

PI3K/AKT, ERK1/2-MAPK and IGF-1R-MAPK signaling pathways[20,21]. *UBQLN4* (Ubiquilin-4, 1q21.3) was shown to regulate Wnt-β-catenin pathway activation in HCC cells[22] and is associated with genomic instability and aggressive tumors[23]. *BYSL* (Bystin Like, 6p21.1), a direct cMYC target[24], is required for nucleologenesis in HCC cell proliferation and its inhibition induces apoptosis, partially arrests the cell cycle and inhibits tumor formation in mouse xenografts[25]. *BYSL* also reportedly promotes glioma/glioblastoma growth via the GSK-3β/β-catenin and AKT/mTOR pathways[26,27].

Of note, there are genes with poorly characterized or controversial roles in HCC tumorigenesis among those that showed strong CNA-mRNA and mRNA-protein correlation. *ATP6V1C1* (ATPase H+ Transporting V1 Subunit C1, 8q22.3) is a member of the vacuolar ATPases (V-ATPases) family of proton pumps with roles in Wnt/β-catenin, Notch, and mTOR signaling, as well as in the regulation of cell invasion, migration and metastasis[28]. The subunit V1C (encoded by *ATP6V1C1*) is, in particular, upregulated in oral cancer, and its silencing impairs breast cancer growth and metastasis[29] but its role in HCC carcinogenesis is unclear. *RRM2B* (Ribonucleotide Reductase Regulatory *TP53* Inducible Subunit M2B, or p53R2, 8q22.3) is a p53 target and a regulator of DNA damage and replication stress[30]. *RRM2B* was reported to be downregulated in HCC and to inhibit cell migration and spreading through the Egr-1/PTEN/Akt1 pathway[31]. Yet its frequent amplification and overexpression in HCC and other cancers suggests it may promote oncogenesis[32,33], specifically in hypoxic conditions[34]. The role of these genes/proteins in HCC may warrant further investigation.

Taken together, our analysis of the CNA-mRNA and mRNA-protein expression correlations showed distinct pathways being regulated on different levels and identified potential HCC driver genes.

**Dysregulated phosphorylation in HCC.** Next, we investigated the protein phosphorylation landscape in HCC. Given that protein phosphorylation may be highly driven by the protein expression rather than changes in phosphorylation, we investigated dysregulated phosphorylation sites with and without normalization by overall protein levels. Differential expression analyses revealed 692 and 648 hyper- and hypophosphorylated sites, respectively, and 302 and 355 normalized (by overall protein levels) hyper- and hypophosphorylated sites compared to normal livers (adjusted $p \le 0.05$ and |log2 fold-change| > 1, Fig. 3a, Supplementary Fig. 9 and Supplementary Data 6). Pathway enrichment analysis revealed that the hyperphosphorylated sites are in proteins involved in cell cycle, mRNA splicing, the immune system, cancer-related signaling pathways such as receptor tyrosine kinases and MAP kinase and regulation by PTEN and p53 (Fig. 3b, Supplementary Fig. 9 and Supplementary Data 7). Signaling by AKT, FGFR, VEGF, TGFβ are also enriched

among the hyperphosphorylated proteins, though not always statistically significant in the analyses of both normalized and unnormalized phosphorylation levels. Among the hypophosphorylated sites, histone modification-related processes were enriched. Interestingly, pathways related to metabolism of amino acids, carbohydrates, lipids were enriched among the overall hypophosphorylated sites but also the normalized hyperphosphorylated sites. By contrast, proteins involved in cellular senescence and chromatin organization showed the opposite trend, with enrichment of overall hyperphosphorylated sites but normalized hypophosphorylated sites. While base excision repair and DNA double-strand break repair were enriched among both hypophosphorylated sites, nucleotide excision repair was enriched among the hyperphosphorylated sites (Fig. 3b, Supplementary Fig. 9 and Supplementary Data 7).

To infer the activation of kinases in HCC, we performed a Kinase-Substrate Enrichment Analysis (KSEA)[35]. KSEA revealed that Aurora kinase A (*AURKA*), Cyclin-dependent kinases 1/2/5/7 (*CDK1/2/5/7*), ERK1/2 (*MAPK1/3*) and PLK1 showed increased activation compared to normal livers, while PKACA/G (*PRKACA/G*), PKCA/Z (PRKCA/Z) and SGK1 showed reduced activity (Fig. 3c and Supplementary Data 8). When analyzing dysregulated phosphorylation normalized by protein level, KSEA revealed decreased AURKA and increased CDK1/2/5, ERK1/2 and GSK3B activity in HCCs (Supplementary Fig. 9 and Supplementary Data 8).

Our results show that altered phosphorylation in HCC affects a wide range of biological processes from cell proliferation and DNA repair to immune system and signal transduction pathways.

**Proteogenomic analysis of significantly mutated genes.** Using whole-exome sequencing, we identified 24,488 somatic mutations (23,660 single nucleotide variants and 828 small insertions and deletions) across the 122 tumor biopsies (Supplementary Fig. 10 and Supplementary Data 9). Using MutSigCV[36] and OncodriveFML[37], we identified 7 significantly mutated genes (SMGs, Supplementary Fig. 10). While *ALB*, *ARID1A*, *AXIN1*, *CDKN1A*, *CTNNB1* and *TP53* had previously been identified as SMGs in several genomic studies[3,38–40], *GPAM* was only identified in a meta-analysis[41] (1.8% vs 7.4% in the current study, Fisher's exact test, $p = 0.001$). Here we found seven of the nine *GPAM* mutations were frameshift mutations, strongly suggestive of a tumor suppressor role (Supplementary Fig. 10). This is corroborated by experiments in HepG2 hepatoblastoma and Huh7 HCC cell lines that knocking down *GPAM* significantly increased cell proliferation (Supplementary Fig. 10).

We evaluated the clinicopathological correlates of the 7 SMGs, together with 6 additional cancer genes identified from at least 2 previous HCC genomics studies[3,36–38] and mutated in ≥3 HCCs of the current cohort (*APOB*, *ARID2*, *CDKN2A*, *KEAP1*, *RB1* and *TSC2*). *TP53* and *CTNNB1* mutations were mutually exclusive

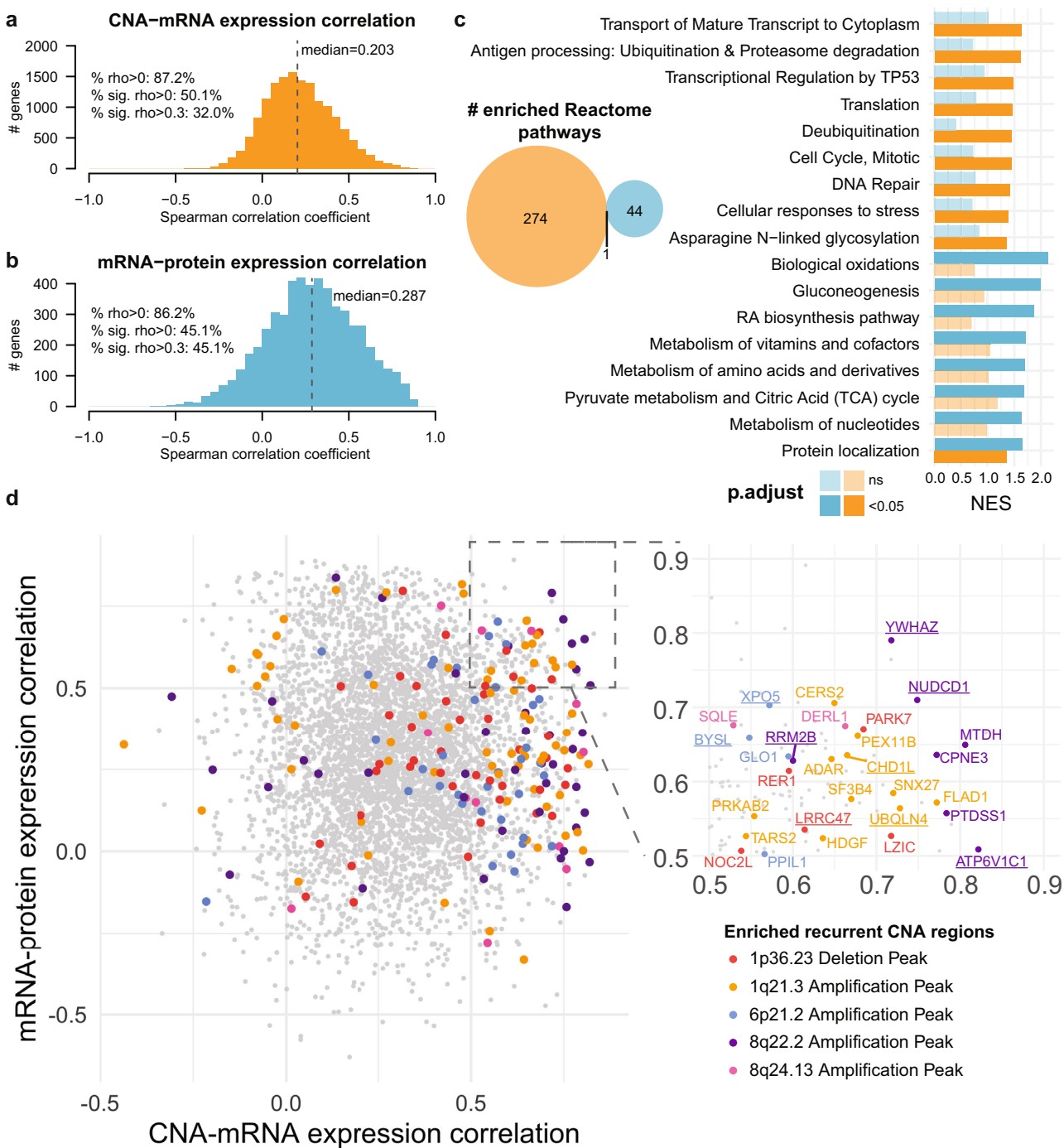

**Fig. 2 CNA-mRNA-protein correlation.** Histograms of the distributions of the per-gene Spearman correlation coefficients (**a**) for CNA-mRNA and (**b**) mRNA-protein expression. sig.: significant. **c** Venn diagram of the number of enriched Reactome pathways for genes/proteins ranked by CNA-mRNA expression correlation (orange) and mRNA-protein expression correlation (blue). Enrichment and statistical significance were defined by gene set enrichment analysis. Multiple correction was performed using the Benjamini–Hochberg method. Barplot of selected Reactome pathways enriched among genes with high CNA-mRNA expression correlation (orange) and/or with high mRNA-protein expression correlation (blue). Statistically significant normalized enrichment scores (NES, $p < 0.05$) are shown in darker shades (dark orange/blue) while non-significant NESs are shown in lighter shades (light orange/blue). **d** Scatterplot of the per-gene Spearman correlation coefficients (y-axis) between mRNA and protein expression against (x-axis) between CNA and mRNA. Genes in five of the recurrently altered regions as defined by GISTIC2 are colored according to the color key. Inset shows the genes with >0.5 correlation coefficients in both comparisons. Dysregulated genes (compared to normal livers) on both the mRNA and protein levels are underlined. Source data are provided as a Source Data file.

($p = 0.012$, Fisher's exact test)[40]. While *CTNNB1*-mutant tumors were more frequently lower grade (Edmondson grade, $p = 0.017$) and were associated with the immune-desert phenotype ($p = 0.039$)[42,43], *TP53*-mutant tumors were of higher grade

($p = 0.001$) and associated with HBV ($p = 0.010$, Supplementary Fig. 10 and Supplementary Data 10). A multivariate Cox-proportional hazard model suggests that *CDKN2A*, *GPAM*, *KEAP1* and *TSC2* mutations are associated with poor overall

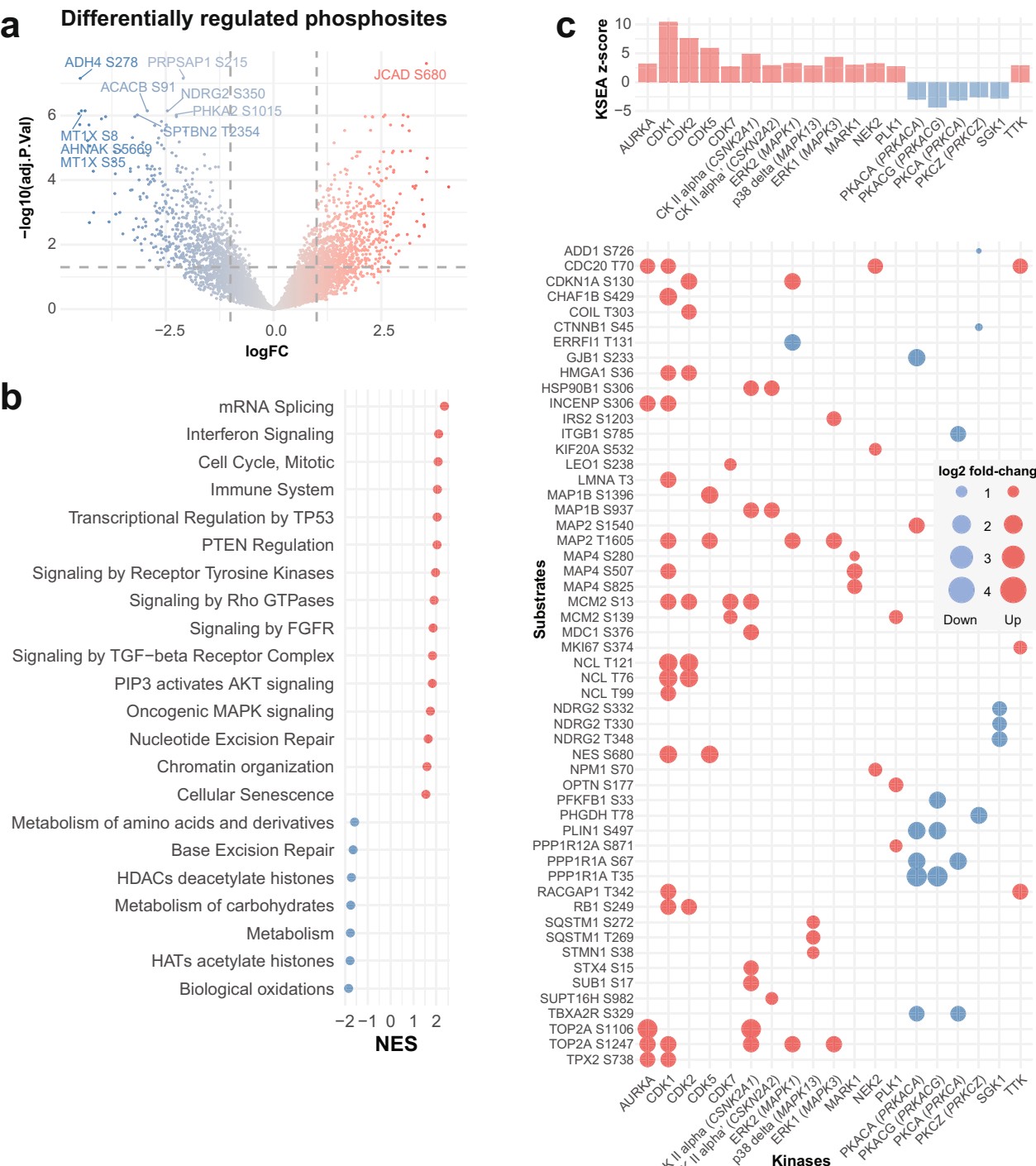

**Fig. 3 The phosphoproteomic landscape of HCC. a** Volcano plot of the −log10(adjusted *p* value) against the log fold-change (logFC) of the differentially regulated phosphorylation sites in HCC compared to normal livers. Dots are colored by logFC. Vertical dotted lines indicate |logFC| = 2 and horizontal dotted line indicates adjusted *p* value = 0.05. **b** Dot plot illustrating selected enriched Reactome pathways according to gene set enrichment analysis (GSEA) from the differential expression analysis in (**a**). NES normalized enrichment score. **c** Top barplot showing the enrichment z-score of the kinases with significantly up- or downregulated kinase activity in a kinase-substrate enrichment analysis (KSEA) comparing HCC to normal livers. In the bubble plot below, the phosphorylation site substrates are shown in rows, where red and blue dots indicate that the phosphorylation site is up- and downregulated, respectively. The size of the dots is proportional to the log2fold-change of the phosphorylation site. Phosphorylation sites with at least a 5-fold difference between HCCs and normal livers are shown. For kinases with <3 substrates with at least a 5-fold difference, the top three substrates with the highest |logFC| are shown. Source data are provided as a Source Data file.

survival independent of BCLC clinical stage (Supplementary Fig. 10 and Supplementary Data 10). An analysis of the SMGs on their cognate mRNA and protein products revealed that *APOB* and *TP53* mutations were associated with lower expression on the

mRNA or the protein levels, while *CTNNB1* mutations were associated with increased expression (Supplementary Fig. 11).

To evaluate the molecular changes associated with *CTNNB1* and *TP53* mutations, we performed differential expression

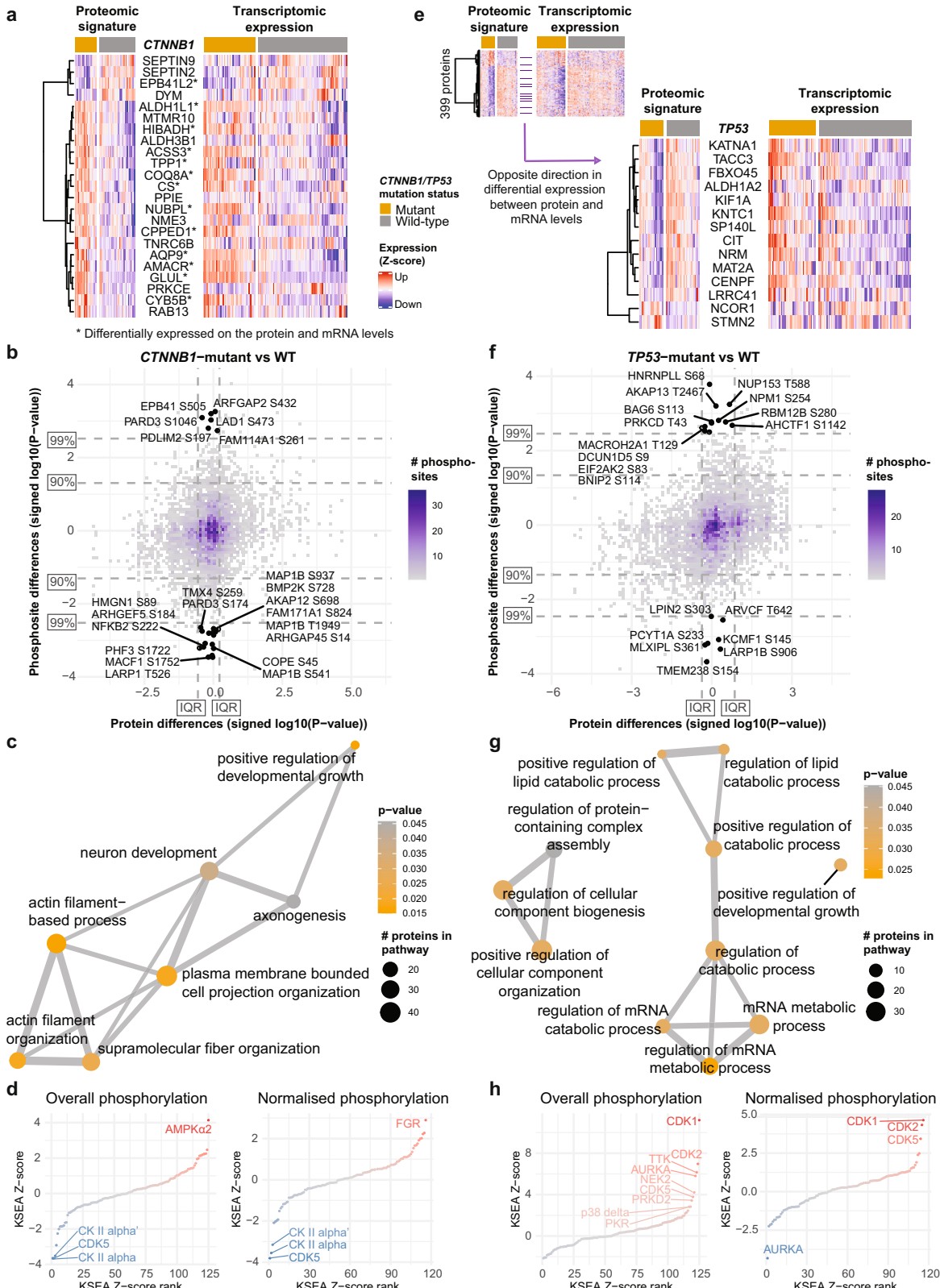

* Differentially expressed on the protein and mRNA levels

analyses comparing mutant and wild-type HCCs. We identified 3067 differentially expressed genes in *CTNNB1*-mutant and 3949 in *TP53*-mutant HCCs, as well as 23 differentially expressed proteins for *CTNNB1*-mutant and 399 for *TP53*-mutant HCCs (Fig. 4a, e and Supplementary Data 11). No statistically significant differences were observed on the phosphoproteome level. Of the 23 proteins that were altered in *CTNNB1*-mutant HCCs, 13 were

also differentially expressed at the mRNA level (Fig. 4a and Supplementary Data 11). These include glutamine synthetase (*GLUL*), α-methylacyl-CoA racemase (*AMACR*, associated with *CTNNB1* mutations in HCC[44]), ACSS3 (*ACSS3*, associated with a metabolic HCC subclass characterized by frequent *CTNNB1* mutations[10]). By contrast, the remaining ten differentially expressed proteins were not associated with differential

**Fig. 4 Proteogenomic analysis of SMGs. a** Heatmaps showing the expression (z-score-transformed) of (left) 23 proteins differentially expressed between *CTNNB1*-mutant and -WT HCCs (FDR < 0.05) and (right) the corresponding gene expression on the mRNA level. Genes with asterisks were also differentially expressed on the transcriptome level. **b** Binned scatterplot plotting the signed (according to the direction of the fold change) *p* values from differential expression analyses of protein expression (x-axis) and of phosphorylation site expression (y-axis) between *CTNNB1*-mutant and -WT HCCs. Phosphorylation sites >99th quantile of the unsigned p-values of the differential phosphoprotein expression analysis and within the inter-quartile range of the signed p-values of differential protein expression analysis are labeled. **c** Enrichment map showing the Gene Ontology biological processes enriched among proteins with phosphorylation sites at >90th quantile of the unsigned p-values of the differential phosphoprotein expression analysis and within the inter-quartile range of signed p-values of differential protein expression analysis. **d** Plot showing the kinase-substrate enrichment analyses z-scores ordered in increasing order, comparing (left) phosphorylation site abundance and (right) phosphorylation site abundance normalized by protein abundance between *CTNNB1*-mutant vs -WT HCCs. Significant kinases are labeled. **e** Small heatmap as (**a**) of (left) 399 proteins differentially expressed between *TP53*-mutant and -WT HCCs (FDR < 0.05) and (right) the corresponding gene expression. Large heatmap showing the subset of 14 proteins/genes for which the direction of the differential expression differed between the proteomic and transcriptomic signatures. **f–h** as (**b–d**) for stratification by *TP53* mutation status. Source data are provided as a Source Data file.

transcription. These include TNRC6B (*TNRC6B*, involved in the β-catenin-independent Wnt signaling), Protein Kinase C Epsilon (*PRKCE*, a β-catenin binding partner[45]), and PPIE (*PPIE*, a spliceosome component that regulates the splicing of the long non-coding RNA *FAST* which in turns regulates β-catenin and Wnt signaling[46]). The Wnt target genes *NKD1, AXIN2, RNF43* and *ALDH3A1*, whose mRNA expression is typically altered in *CTNNB1*-mutant HCCs, are not differentially expressed at the protein level (Fig. 4a and Supplementary Data 11).

*CTNNB1* encodes β-catenin, a protein involved in intercellular adhesion. In HCC, *CTNNB1* mutations lead to the accumulation of cytoplasmic β-catenin and the subsequent aberrant Wnt activation. We searched for phosphorylations that differ the most between *CTNNB1*-mutant and *CTNNB1*-wild-type HCCs (10% most extreme p-values from differential expression analysis) but are not associated with differences on the protein level (interquartile range of p-value signed according to the direction of differential expression, Fig. 4b). Pathway analysis showed that 189 such phosphorylation sites are in proteins involved in actin filament organization (Fig. 4c and Supplementary Data 12). Among the hyperphosphorylated sites in *CTNNB1*-mutant HCCs were Par3-alpha (encoded by *PARD3*) S1046 and PDLIM2 (*PDLIM2*) S197, while hypophosphorylation was observed in MAP1B (*MAP1B*) S541/S937/S1396 and ACF7 (*MACF1*) S1752. Phosphorylation or loss of Par3-alpha has been linked to the loss of cell polarity[47,48]. Similarly, PDLIM2 phosphorylation leads to its stabilization and facilitates β-catenin activation[49]. By contrast, Wnt activation inactivates GSK3 kinase and causes decreased MAP1B and ACF7 phosphorylation, resulting in increased microtubule stability[50] and migration[51,52]. Indeed, we observed reduced expression of the epithelial markers *CDH1* (E-cadherin) and *KRT19* (Keratin-19) in the *CTNNB1*-mutant HCCs, suggesting a loss of the epithelial phenotype (Supplementary Fig. 12). Moreover, KSEA revealed that, compared to *CTNNB1*-wild-type HCCs, *CTNNB1*-mutant HCCs showed increased AMPKα2 kinase activity and reduced activity of CK II alpha, CK II alpha' and CDK5 (Fig. 4d). Instead, KSEA of the normalized phosphorylation levels showed increased activity of FGR, a kinase that contributes to the regulation of immune response and cytoskeleton remodeling (Fig. 4d).

Similarly, of the 399 differentially expressed proteins between *TP53*-mutant and -wild-type HCCs, 238 were differentially expressed at the mRNA level (Supplementary Data 11). While stathmin 2 (*STMN2*) and Nuclear receptor corepressor 1 (*NCOR1*) were overexpressed at the protein level, they were underexpressed on the mRNA level (Fig. 4e). By contrast, another 12 were underexpressed on the protein level but overexpressed on the mRNA level, and these included Centromere Protein F (*CENPF*), TACC3 (*TACC3*) and Kinetochore-associated protein 1 (*KNTC1*), all involved in the regulation of the mitotic spindle. We further identified 178 sites whose phosphorylation differed

between *TP53*-mutant and -wild-type HCCs without differences on the protein level (Fig. 4f). A pathway analysis of these phosphorylation sites suggests that *TP53* mutations are associated with phosphorylation changes in proteins involved in the regulation of lipid and mRNA metabolic processes, and the regulation of cellular component biogenesis and organization (Fig. 4g and Supplementary Data 12). In particular, PKR (*EIF2AK2*) S83 is an activating autophosphorylation site[53]. PKR is involved in diverse cellular processes, including stress response against pathogens (e.g., HCV) and its activation inhibits protein synthesis[54]. HCV infection triggers PKR phosphorylation[55], though here we did not observe an enrichment of HCV-associated HCCs among the *TP53*-mutant HCCs ($p > 0.05$, Fisher's exact test). KSEA revealed that *TP53*-mutant HCCs showed increased CDK1/2/5 activity (Fig. 4h). Other kinases involved in the control of cell cycle, mitotic checkpoint and spindle formation (Aurora Kinase A, TTK, NEK2), protein synthesis and stress response (PKR), and MAPK signaling (PRKD2, p38 delta (*MAPK13*)) also showed increased activity when comparing overall phosphorylation levels.

Taken together, the proteogenomic analysis suggests that the loss of epithelial phenotype seen in *CTNNB1*-mutant HCCs may result from alterations in phosphorylation of proteins involved in the organization of actin filaments, thereby regulating cell polarity and migration, whereas *TP53*-mutant HCCs are associated with altered phosphorylation of proteins related to lipid metabolism and cell cycle control.

**Molecular subtypes of HCC.** We performed unsupervised analyses to identify HCC subtypes in each of the 5 individual omics data sets, namely, somatic mutation, CNA, transcriptome, proteome and phosphoproteome ('single-omics') as well as an integrative analysis incorporating all 5 ($n = 51$ with complete data, Supplementary Fig. 13). For the single-omics analyses, we identified 2–4 robust clusters using two independent approaches. The four somatic mutation subclasses were characterized by mutations in *CTNNB1*, *TP53*, or *ARID1A* or the lack of *CTNNB1*/*TP53*/*ARID1A* mutations, the two CNA subclasses were distinguished by the level of genomic instability, while the three transcriptome subclasses were distinguished by elevated cell cycle related processes, immune pathways or metabolism (Supplementary Fig. 13 and Supplementary Data 13). Of note, mutation cluster 3 and transcriptome cluster 1 were associated with high Edmondson grade. Similarly, the mutation and transcriptome clusters but not the CNA subclasses were associated with the Hoshida molecular subtypes[6] (Supplementary Fig. 13).

For the proteome and phosphoproteome, in each case two clusters were identified. Proteome subclass 1 is associated with ribonucleoprotein organization, and mRNA splicing and processing, while subclass 2 is associated with metabolism and

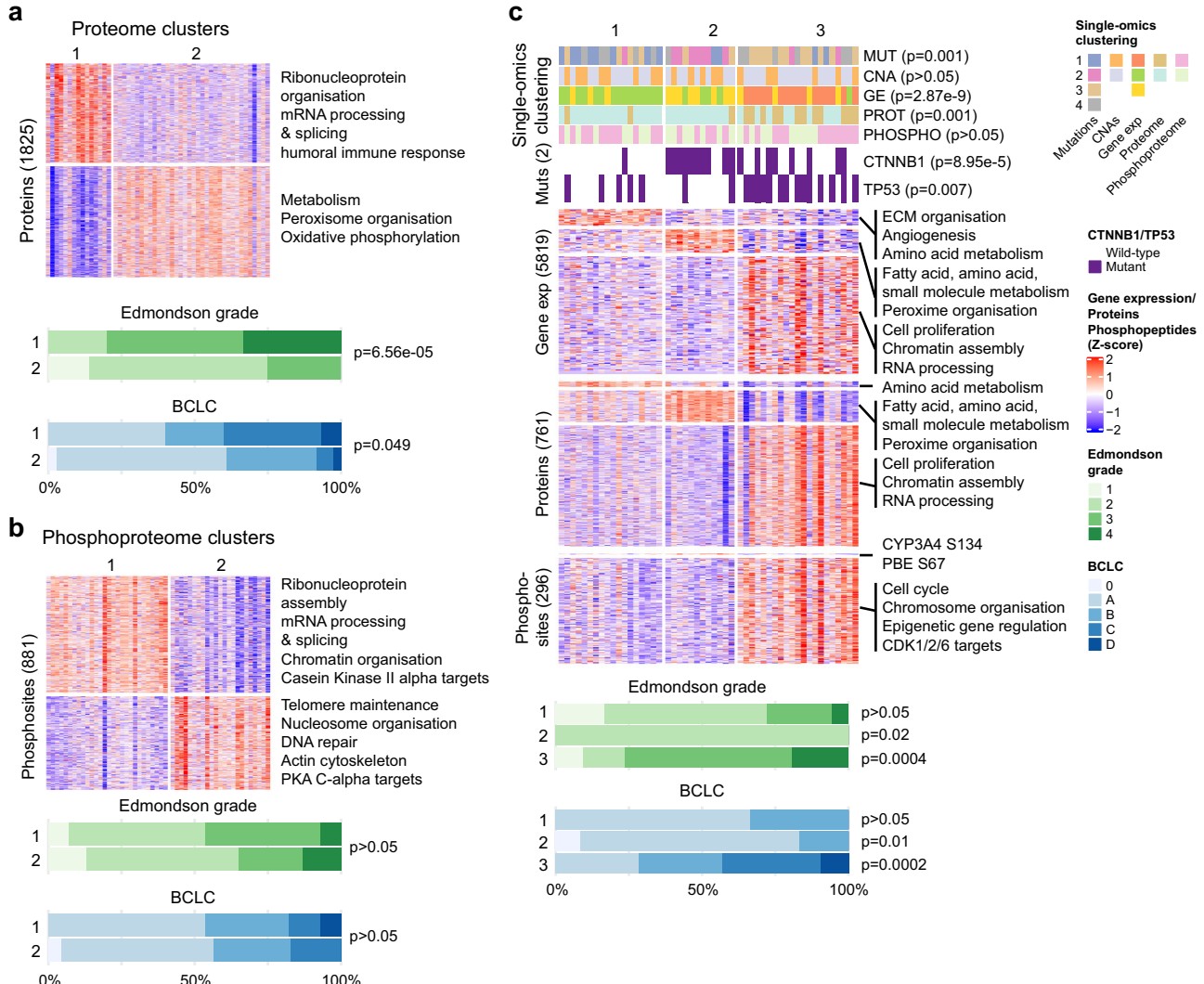

**Fig. 5 Integrated phosphoproteomic classification of HCC.** Unsupervised clustering of the (**a**) proteome and (**b**) phosphoproteome data using consensus non-negative matrix factorization. **c** Integrative clustering of the mutation, copy number alteration, transcriptome, proteome and phosphoproteome using the iCluster method. Copy number alterations not shown in the figure as no genomic region differed between clusters. **a–c** Barplots below show the distribution of Edmondson grade and BCLC between the clusters. Statistical comparison for each cluster was computed using the two-sided Mann–Whitney *U* test. Source data are provided as a Source Data file.

oxidative phosphorylation (Fig. 5a and Supplementary Data 13). Phosphoproteome subclass 1 is also linked to ribonucleoprotein assembly and mRNA splicing and processing but also to chromatin organization and CK II alpha activation. Instead, phosphoproteome subclass 2 is associated with telomere maintenance, nucleosome organization, DNA repair, actin cytoskeleton regulation and PKA C-alpha activation (Fig. 5b and Supplementary Data 13). The proteome clusters, but not the phosphoproteome clusters, are associated with Edmondson grade and the Hoshida molecular subtypes[6] (Fig. 5a, b and Supplementary Fig. 13). In particular, proteome cluster 1 was enriched for high-grade and high-stage HCCs. Further, significant association was observed between mutation, transcriptome and proteome clusterings, but not with the phosphoproteome clusters (Supplementary Fig. 13).

We then asked whether an integrative clustering using all five omics would provide further insight into the diversity of HCC. Using the iCluster method[56], we defined three robust clusters (Fig. 5c), which are largely recapitulated using the algorithmically

distinct similarity network fusion (SNF) approach[57] (Supplementary Fig. 13). To investigate how the three clusters differ from each other, we compared them on the individual molecular levels (Supplementary Data 14). In cluster 1, we observed an enrichment of genes involved in ECM organization, angiogenesis and amino acid metabolism on the mRNA level and amino acid metabolism on the protein level. In cluster 2, pathways related to fatty acid, amino acid and small molecule metabolism were enriched on both the mRNA and protein levels. In cluster 3, pathways related to cell proliferation, chromatin assembly and RNA processing were enriched on both the mRNA and protein levels, while on the phosphoprotein level, we observed pathways related to cell cycle, epigenetic gene regulation and activation of CDK1/2/6. No specific CNAs were enriched in any subclass. Integrative clusters 2 and 3 were associated with low- and high-grade HCCs, respectively. Compared to the single-omics clustering, the integrative clusters were associated with the mutation, transcriptome and proteome clusters but not with the CNA or the phosphoproteome clusters (Fig. 5c). Furthermore, we also found

that the integrative clusters, as well as the single-omics clustering of mutation, transcriptome and proteome to be associated with the previously published molecular subtypes defined by Hoshida et al.[6] (Supplementary Fig. 13).

Finally, to assess whether the single-omics and integrative molecular subclasses are prognostic, we performed univariate and multivariate Cox proportional hazard analyses. In the univariate analyses, transcriptome cluster 1 (increased cell proliferation, $p = 0.037$), proteome cluster 1 (ribonucleoprotein organization and mRNA processing, $p = 0.022$) and integrative cluster 3 (increased cell proliferation, epigenetic gene regulation, and *TP53* mutation, $p = 0.005$) were associated with poor overall survival, while proteome cluster 2 (metabolism, $p = 0.022$) and integrative cluster 2 (metabolism and *CTNNB1* mutation, $p = 0.015$) were associated with improved overall survival (Supplementary Table 1). In the multivariate analyses incorporating primary prognostic indicator BCLC, mutation cluster 2 (*CTNNB1*-mutant) was associated with good prognosis ($p = 0.025$) and mutation cluster 3 (*TP53*-mutant) was associated with poor prognosis ($p = 0.034$) independent of BCLC (Supplementary Table 1).

In summary, while molecular clustering of the proteome data largely recapitulated that of the transcriptome and histological grading, molecular clustering of the phosphoproteome data differed from that of other single-omics and integrative clustering.

## Discussion

Our integrated proteogenomic analysis of HCC biopsies across etiologies and clinical stages revealed similar but also distinct biological processes, metabolic reprogramming and signaling pathway activation on the different molecular levels. We found that RNA processing was consistently upregulated and metabolic pathways were downregulated on the transcriptome, proteome and phosphoproteome levels. Interestingly, translational regulation-related genes were upregulated on the mRNA but not the protein level. One could speculate that the increased transcription of translational regulation-related genes may be a compensatory mechanism for increased protein degradation. On the phosphoproteome level, altered protein phosphorylation was associated with pathways related to cell cycle, immune system, and DNA repair, as well as oncogenic signaling pathways such as MAPK, PI3K/Akt/PTEN and FGFR. Indeed, KSEA revealed the increased activity of ERK1/2 and cell cycle-related kinases such as AURKA, CDK1/2/5/7, PLK1 and TTK.

We also identified putative HCC driver genes and drug targets. We showed that *GPAM* is a frequently mutated gene in HCC. Among genes and proteins that show positive correlation on the CNA-mRNA and mRNA-protein levels and are dysregulated in HCC, we identified candidate driver genes, such as *NUDCD1*, *UBQLN4*, *BYSL*, *ATP6V1C1* and *RRM2B* involved in diverse processes including EMT, cell cycle and DNA damage regulation, and the regulation of the Wnt-β-catenin, AKT/mTOR and Notch pathways. At least two of them have been suggested as therapeutic targets. Specifically, *UBQLN4* is reported to repress homologous recombination repair and promotes sensitivity to PARP1 inhibition[23], while *RRB2M*, a gene that appears to promote tumorigenesis in hypoxic conditions and encodes a component of the ribonucleotide reductase, may be targeted by gemcitabine[34], a chemotherapeutic agent sometimes used to treat HCC. Our phosphoproteome analysis also revealed targetable kinases with elevated activity, especially Aurora Kinase A and CDKs. Aurora Kinase A and CDK1/2 are classical cell cycle-related kinases whereas CDK5 regulates many biological processes, including angiogenesis and DNA damage response[58,59]. Preclinical studies

have shown that Aurora Kinase A and CDK1/2/5 inhibitors are efficacious in HCC models[60–62] and may act synergistically with sorafenib/regorafenib and chemotherapeutic agents[59,63,64]. Integrative clustering of HCCs reveal that cluster 2 was enriched for *CTNNB1* mutants, suggesting Wnt signaling may be a potential therapeutic target for tumors in this cluster. On the other hand, integrative cluster 3 is characterized by elevated cell cycle signatures and CDK1/2/6 activity. CDKs, in particular CDK4/6, are well-established therapeutic targets. These results also suggest potential molecularly driven putative drug targets in specific HCC subsets.

*CTNNB1* and *TP53* are the two most frequently mutated genes in HCC. We identified altered phosphorylation sites in proteins, such as ACF7, MAP1B, PDLIM2 and Par3-alpha, that may underpin the loss of epithelial phenotype frequently seen in *CTNNB1*-mutant HCCs. ACF7 (*MACF1*), part of the β-catenin destruction complex[65], is required for stabilizing Axin upon Wnt activation[66]. Wnt activation causes GSK3 inactivation and inability to phosphorylate ACF7, and dephosphorylated ACF7 remains active and able to form necessary connections between microtubules and the actin cytoskeleton to enable migration[51,52]. Similarly, GSK3 inactivation decreases MAP1B phosphorylation and facilitates microtubule assembly and migration[50]. By contrast, PDLIM2 is required for polarized cell migration and its phosphorylation facilitates β-catenin activation and nuclear translocation[49]. Par3-alpha (*PARD3*), a regulator of tight junction assembly[67], changes the affinity for its interaction partners[68–70] upon phosphorylation, leading to loss of cell polarity and induction of migration. KSEA of *CTNNB1*-mutant HCCs also revealed increased activity of FGR, involved in immune response and cytoskeleton remodeling. In contrast to a previous study of HBV-associated HCCs[12], we did not observe elevated ALDOA S36 phosphorylation in our cohort of *CTNNB1*-mutant HCCs (FDR = 0.98). Instead, KSEA of *TP53*-mutant HCCs identified increased activity of CDK1/2/5, which could in part explain the higher histological grade typically associated with *TP53* mutations. Our analysis also identified altered phosphorylation of proteins involved in lipid metabolism in *TP53*-mutant HCCs. p53 with a gain-of-function mutation has been reported to promote lipid synthesis via at least two mechanisms, by activating the SREBP transcription factors and by inhibiting AMPK[71,72]. Finally, as previously reported[13], the presence of *TP53* mutations is associated with loss of expression on the mRNA level.

Proteome and phosphoproteome classifications revealed two clusters each. Although for both classifications, one of the clusters was associated with overexpression or enhanced phosphorylation of proteins involved in ribonucleoprotein organization, and mRNA processing and splicing, there was little concordance between the two classifications. The lack of concordance of the phosphoproteome clusters was also seen with mutation, transcriptome and integrative clusters. As protein phosphorylation is highly dynamic and our profiling captures a snapshot of the tumor, phosphoproteomic data are inherently noisier than other types of molecular data. Integrative clustering, using the algorithmically distinct iCluster and SNF, identified three clusters that resembled the spectrum of Edmondson grade and BCLC. They also resemble the three proteome subclasses identified in two previous studies in HBV-related HCC, both of which described subclasses characterized by metabolic reprogramming, microenvironment dysregulation and cell proliferation[11,12]. In particular, the proliferation proteome subclasses in HBV-related HCC were associated with tumor thrombus[12] and microscopic vascular invasion[11]. Here our integrative cluster 3 (increased cell proliferation and *TP53* mutation) is associated with macro-vascular invasion ($p = 0.01$, Fisher's exact test) and poor outcomes. However, it should be noted that our cohort was accrued over a

long period and clinical practice has changed significantly over the past decade, hence outcome data are inherently difficult to interpret.

Proteomic and phosphoproteomic profiling technologies have improved significantly over the past decade but are not without challenges. In our study, we performed (phospho)proteomic profiling for about half of the samples but in triplicates to ensure robust statistics. Compared to WES and RNA-seq, (phospho) proteome data are relatively sparse and not all proteins and phosphorylation sites can be detected. It should also be noted that not all proteins are expressed at a given time at detectable levels and the detection of phosphorylation sites is challenging due to the high degree of variability between tissues and samples. Further, some phosphorylation sites may never be detected as they are on hard-to-detect peptides. Our study has also generated interesting leads for future studies. Mutagenesis experiments would be required to properly dissect the effect of *CTNNB1* and *TP53* mutations on the phosphoproteome. The fact that seven of the nine *GPAM* mutations were frameshift mutations together with the observation that knockdown of *GPAM* significantly increased cell proliferation strongly suggests a tumor suppressor role for *GPAM* in HCC. A full characterization of the functional role of the individual frameshift mutations in *GPAM* would require further experiments.

Our study generated proteome and phosphoproteome data (Supplementary Figs. 2 and 3) for a wide spectrum of HCCs across all clinical stages and major etiologies, thus more representative of the molecular heterogeneity of HCC than previous studies[11–13]. In particular, our analysis of the phosphoproteome has provided deeper insights into the major processes and kinases altered in HCC compared to previous studies. Our analyses of the CNA-mRNA-protein correlation, phosphoproteome and integrative clusters of HCCs have also nominated a number of putative drivers and drug targets that may warrant further study in the future. In conclusion, our study provides a comprehensive analysis of the proteomic and phosphoproteomic landscape of HCCs, identifying proteome and phosphoproteome alterations underlying HCC.

## Methods

**HCC biopsy procedure and sample collection**. Human tissues were obtained from patients undergoing diagnostic liver biopsy at the University Hospital Basel between 2008 and 2018. Written informed consent was obtained from all patients. The study was approved by the ethics committee of the northwestern part of Switzerland (Protocol Number EKNZ 2014-099). Ultrasound-guided needle biopsies were obtained from tumor lesion(s) and the liver parenchyma at a site distant from the tumor with a coaxial liver biopsy technique that allows taking several biopsy samples through a single biopsy needle tract as described[73]. Clinical disease staging was performed using the Barcelona Clinic Liver Cancer (BCLC) system[74]. Biopsies from multicentric tumors (i.e. genetically independent primary tumors), but not intra-hepatic metastases, were included. In total, 122 HCC biopsies and 115 non-tumoral tissues from 114 patients were included in the study (Supplementary Data 1), including 6 patients from whom 2 synchronous multicentric tumor biopsies and 1 patient from whom 3 multicentric tumor biopsies were obtained (Supplementary Fig. 1). None of the patients had received systemic or locoregional therapies for liver cancer prior to biopsy. Two patients were treated with curative surgery or ablation and were biopsied after HCC recurrence was diagnosed by imaging.

As control, we used liver biopsies with normal histology obtained from 19 patients without HCC and with normal liver values (Supplementary Data 1). The biopsy procedure was as described above.

**Histopathological assessment**. Diagnosis of HCC and histopathology evaluation were carried out on FFPE slides blindly by at least two board-certified hepato-pathologists (CE, MSM and/or LMT) at the Institute of Pathology of the University Hospital Basel. Histopathologic grading was performed according to the Edmondson grading system[73,75]. Hematoxylin & eosin (H&E) slides were reviewed to define the presence or absence of cirrhosis, underlying liver disease, cholestasis, vessel infiltration, necrotic areas, major growth pattern, cytological variants and special subtypes according to the guidelines by the World Health Organization[76]. Immunophenotypes were classified according to Chen and Mellman[77]. Specifically, inflamed tumors are defined as tumors in which tumor infiltrating lymphocytes (TILs) are present in the tumor parenchyma in close proximity to tumor cells; immune-excluded are tumors in which TILs are present only in ≥10% of the tumor stroma and/or tumor margins is populated by lymphocytes located in the immediate vicinity of tumor cells; and immune-desert, in which less than 10% of the tumor stroma is populated by lymphocytes, and neither dense immune cell infiltrates nor immune cells are in contact with tumor cells.

**DNA and RNA extraction**. Genomic DNA and total RNA from tumor and adjacent liver parenchyma were extracted using the ZR-Duet DNA and RNA MiniPrep Plus kit (Zymo Research) following the manufacturer's instructions. Prior to extraction, biopsies were crushed in liquid nitrogen to facilitate lysis. Total RNA of 15 patients without HCC was extracted using Trizol (Thermo Fisher Scientific) according to the manufacturer's instructions. Extracted DNA was quantified using the Qubit Fluorometer (Invitrogen). Extracted RNA was quantified using NanoDrop 2000 spectrophotometer (Thermo Fisher Scientific), and RNA quality/integrity was assessed with an Agilent 2100 BioAnalyzer using RNA 6000 Nano Kit (Agilent Technologies).

**Whole-exome sequencing and data processing**. All 122 HCC biopsies and 115 non-tumoral tissues from 114 patients were subjected to whole-exome sequencing. Whole-exome capture was performed using the SureSelectXT Clinical Research Exome (Agilent Technologies) or SureSelect Human All Exon V6 + COSMIC (Agilent Technologies) platforms according to the manufacturer's guidelines. Sequencing was performed on an Illumina HiSeq 2500 at the Genomics Facility Basel according to the manufacturer's guidelines. Paired-end 101-bp reads were generated. Tumor biopsies and non-tumoral biopsies were sequenced to median depths of 94.3 (range 16.4–140.0) and 49.4 (range 34.5–86.2), respectively (Supplementary Data 1).

Sequence reads were aligned to the reference human genome GRCh37 using Burrows-Wheeler Aligner (BWA, v0.7.12/13)[78]. Local realignment, duplicate removal and base quality adjustment were performed using the Genome Analysis Toolkit (GATK, v3.6)[79] and Picard (http://broadinstitute.github.io/picard/, v2.4.1). Somatic single nucleotide variants (SNVs) and small insertions and deletions (indels) were detected using MuTect (v1.1.4)[80] and Strelka (v1.0.15)[81], respectively. We filtered out SNVs and indels outside of the target regions, those with variant allelic fraction (VAF) of <1% and/or those supported by <3 reads. We excluded variants for which the tumor VAF was <5 times that of the paired non-tumor VAF. We further excluded variants identified in at least two of a panel of 123 non-tumor samples, including the 115 non-tumor samples included in the current study, captured and sequenced using the same protocols using the artifact detection mode of MuTect2 implemented in GATK 3.6. Mutations affecting hotspot residues[82] were annotated. Significantly mutated genes were identified using MutsigCV (v1.4)[36] and OncodriveFML (accessed 08-07-2020). Genes with $q < 0.1$ were considered significantly mutated. 'Lollipop' plots were generated using the 'MutationMapper' tool (accessed 12-1-2019) on the cBioPortal[83]. Mutual exclusivity and co-occurrence of significantly mutated genes were computed using one-sided Fisher's exact test ($p < 0.05$), where log2 odds ratio >0 indicates occurrence and log2 odds ratio <0 indicates mutual exclusivity. Tumor mutational burden was defined as the total number of somatic mutations (including synonymous and nonsynonymous point mutations and indels) in the coding region and splice sites.

Allele-specific CNAs were identified using FACETS (v0.5.5)[84], which performs a joint segmentation of the total and allelic copy ratio and infers allele-specific copy number states. Somatic mutations associated with the loss of the wild-type allele (i.e., loss of heterozygosity [LOH]) were identified as those for which the lesser (minor) copy number state at the locus was 0. All mutations on chromosome X in male patients were considered to be associated with LOH. Copy number states were collapsed to the gene level based on the median values to coding gene resolution based on all coding genes retrieved from the Ensembl (release GRCh37.p13). Genes with total copy number greater than gene-level median ploidy were considered gains; greater than ploidy + 4, amplifications; less than ploidy, losses; and total copy number of 0, homozygous deletions. Fraction of genome altered was computed as the fraction of genes with amplification, gain, loss or deletion. Tumors with >5% of the genome at copy number 0 (homozygous deletions, 5 tumors) were excluded from the identification of homozygous deletions and from the computation of fraction of genome altered. Significant focal CNAs were identified from segmented data for all 122 tumor biopsies using GISTIC2.0 (v2.0.23)[85].

**RNA-sequencing and data processing**. RNA-seq library prep was performed with 200 ng total RNA using the TruSeq Stranded Total RNA Library Prep Kit with Ribo-Zero Gold (Illumina) according to manufacturer's specifications. Single-end 126-bp sequencing was performed on an Illumina HiSeq 2500 using v4 SBS chemistry at the Genomics Facility Basel according to the manufacturer's guidelines. Primary data analysis was performed with the Illumina RTA version 1.18.66.3.

Sequence reads were aligned simultaneously to the human reference genome GRCh37, HBV strain ayw genome (NC 003977.2), and HCV genotype 1 genome

(NC 004102.1) by STAR (v2.5.2a)[86] using the two-pass approach. Median numbers of reads aligning to the human genome were 52.2 million (range 37.4–115.1 million) and 63.5 million (range 52.5 - 82.2 million) for the HCC and normal liver biopsies, respectively (Supplementary Data 1).

Transcript quantification was performed using RSEM (v1.2.31)[87]. Gene-level expected counts were upper-quartile-normalized to 1000. For downstream analysis, we computed the log2-fold-changes of normalized RSEM gene counts between tumors and the median of 15 normal livers. Molecular subtyping according to Hoshida et al.[6] was performed using the Nearest Template Prediction (http://software.broadinstitute.org/cancer/software/genepattern, accessed 29-06-2018).

**Biopsy sample preparation for proteomics, protein extraction and digestion**. Fresh liver biopsies from 51 HCC and 11 normal livers were immediately snap-frozen in liquid nitrogen. The average time from removing the biopsy from the liver to freezing took about 2 min. The biopsies were processed as previously described[88]. Specifically, for protein extraction, each frozen biopsy was crushed in an in-house constructed metal mortar cooled on dry ice into a fine powder (cryogenic grinding) and transferred into a cooled 1.5 ml tube containing 150–400 ml lysis buffer (50 mM Tris-HCl pH 8.0, 8 M urea, 150 mM NaCl, 1 mM PMSF, Complete Mini Protease Inhibitors (Sigma-Aldrich), PhosSTOP Phosphatase Inhibitors (Sigma-Aldrich)). The biopsy lysate was vigorously vortexed for 5 min, rotated for 1 h at 4 °C and sonicated twice in a VWR Ultrasonic cleaner bath (USC300T) for 1 min. Next, the lysate was centrifuged for 10 min at 15 °C at 20,000 × g and supernatant was removed and stored. Next, 50 µl of fresh lysis buffer were added and the sample was homogenized with a Teflon pestle in a hand homogenizer (Pellet Pestle Motor, Kontes/Kimble, USA) at maximum speed on ice twice for 1 min. Samples were centrifuged for 10 min at 15 °C at 20,000 × g and supernatant was removed and pooled with the previous one. Protein concentration was measured with a Bradford assay. Next, proteins were reduced with 10 mM DTT for 1 h at 37 °C and alkylated with 50 mM iodoacetamide for 30 min at RT in the dark, both with gentle shaking. Urea concentration was lowered to 4 M with 50 mM Tris-HCl, pH 8.0. Lysates were digested with two rounds of endoproteinase LysC (Wako) at a 1:100 enzyme-to-protein ratio at 37 °C for two hours. Next, the urea concentration was lowered to 1 M. Lysates were digested with two rounds of trypsin (Sigma): 1:50 ratio overnight and 1:100 ratio for 2 h at 37 °C. Digestion was stopped with TFA to a final concentration of 0.5%. Digests were centrifuged for 2 min at 1500 × g and desalted on a C18 SepPak cartridge (50 mg column for up to 2.5 mg peptide load capacity, Waters) or C18 Macrospin/Microspin cartridge (Waters). Peptide concentration was estimated at 280 nm, aliquots were taken and peptides were dried in the SpeedVac and stored at −20 °C.

**Peptide fractionation for proteome of human HCC biopsies**. Human HCC biopsies were measured by sequential window acquisition of all theoretical mass spectra (SWATH), in which data-independent acquisition is coupled with spectral library match[14]. From each biopsy, 30 µg of peptides were used for SWATH analysis and 100 µg of peptides were used for library preparation. The biopsies from the 11 patients with healthy livers were measured individually and also as a pool. This pool was measured as a reference several times over the course of the project to account for potential batch effects. Ten biopsy samples were measured together as one batch of samples on the same capillary column. For library generation, 100 µg of peptides from each of the 10 biopsies of one batch were pooled together and subjected to high-pH fractionation with a total of 1 mg of peptide injected by 5 individual injections of 200 µg. Peptides were fractionated by high-pH reversed phase separation using a XBridge Peptide BEH C18 column (3.5 µm, 130 Å, 1 mm × 150 mm, Waters) on an Agilent 1260 Infinity HPLC system. Peptides were loaded on column in buffer A (ammonium formate (20 mM, pH 10) in H2O) and eluted using a two-step linear 60 min gradient from 2% to 50% (v/v) buffer B (90% acetonitrile/10% ammonium formate (20 mM, pH 10) at a flow rate of 42 µl/min. Elution of peptides was monitored with a UV detector (215 nm, 254 nm). A total of 36 fractions were collected and subsequently pooled into 12 fractions using a post-concatenation strategy as previously described[89] by combining fractions 1, 13, 25; 2, 14, 26; and so on. Peptides were dried in a SpeedVac, resuspended in 0.1% formic acid (mobile phase A) and OD was measured. Twenty µg of each fraction were used for library measurements.

**SWATH analysis and library preparation**. The biopsy samples were analyzed on a Thermo Fisher QExactive Plus instrument coupled to an Easy nLC 1000. For SWATH analysis of the biopsy samples, 1.1 µg was injected on column including 10% of iRT peptide mix (HRM kit Ki-3003, Biognosys, Zurich, Switzerland). For library generation, 2 µg of each high pH fraction including 10% of iRT peptide mix (HRM kit Ki-3003) were injected on column. Proteomes were analyzed by capillary LC/MS/MS using a homemade separating column (0.075 mm × 38 cm) packed with Reprosil C18 reverse-phase material (2.4-µm particle size; Dr. Maisch). The solvents used for peptide separation were 0.1% formic acid (solvent A) and 0.1% formic acid and 80% AcCN in water (solvent B). Two microliters of sample were injected. A linear gradient from 0–40% solvent B in solvent A in 190 min was delivered with the nano pump at a flow rate of 200 nL/min. After 190 min, solvent B was increased to 95% in 5 min. The eluting peptides were ionized at 2.5 kV. Singly charged ions and ions with unassigned charge state were excluded from

triggering MS2 events. For SWATH measurements, one Full MS-SIM scan at resolution of 70,000 was followed by 40 mass windows of dynamic size ranging from 400 to 1600 m/z with 4 kDa overlap at a resolution of 17,500. For library measurements, the mass spectrometer was operated in data-dependent mode and the precursor scan was done in the Orbitrap at 70,000 resolution. A top-20 method was run. For SWATH analysis and library generation, samples were injected in triplicates.

**SWATH data analysis**. The library was generated with MaxQuant (version 1.5.1.2)[90] using the default settings except that the mass tolerance of the instrument was set to 10 ppm and the minimal ratio count for quantification was set to 1. The Uniprot SwissProt database (17 August 2015) including the iRT fusion peptide was used for the searches. All library measurements were pooled into one MaxQuant analysis to generate one general HCC library. The raw SWATH MS runs of the individual biopsies were converted using the HTRMS converter (Biognosys). The converted SWATH runs were analyzed with Spectronaut X (Version 12.0.20491.20.29183) (Biognosys) using the default settings and searched against our in-house generated general HCC library.

**Proteome analysis**. Raw protein-group based data were exported from Spectronaut and imported into FileMakerPro Advanced (Version FMP18) for further data processing. The raw intensities of the triplicates were averaged and the mean values transformed by the logarithm to the base 2. Next, the values were normalized by median subtraction. To account for potential batch effects, the log2 median-subtracted intensities of each biopsy were normalized to the mean intensity of all measured runs of the pool of healthy liver tissue. The proteome of the patient biopsies was continuously measured over a time frame of 2 years. Throughout this time period also aliquots of the pooled healthy sample were measured. All measured runs of the pooled healthy sample were therefore averaged for normalization.

We obtained data for 6167 proteins that were quantified (always against healthy liver) in at least one run in at least one HCC (Supplementary Fig. 2 and Supplementary Data 1), 5612 proteins quantified in at least 26 HCCs and 1997 in all 51 HCCs. Starting with the 6167 proteins quantified in at least one HCC, we removed proteins for which data were missing from >50% of the samples (50% to enable sufficient data for imputation), resulting in 5631 proteins for further analysis. Data imputation using nearest neighbor averaging was performed using the 'impute.knn' function from the 'impute' R package (v1.64.0).

**Phospho-proteome analysis**. Peptide samples were enriched for phosphorylated peptides using Fe(III)-IMAC cartridges on an AssayMAP Bravo platform as recently described[91]. Specifically, phosphorylated peptides were enriched using Fe(III)-NTA 5 µL cartridges (Agilent technologies) in an automated fashion using the AssayMAP Bravo Platform (Agilent Technologies). Fe(III)-NTA cartridges were primed with 250 µL of 0.1% TFA in ACN and equilibrated with 250 µL of loading buffer (80% ACN/0.1% TFA). Samples were dissolved in 160 µL of loading buffer and loaded onto the cartridge. The columns were washed with 250 µL of loading buffer, and the phosphorylated peptides were eluted with 25 µL of 1% ammonia directly into 25 µL of 10% formic acid. Samples were dried in the speedvac at low temperature and stored at −20 °C. We used an input peptide amount of approx. 165 µg for the phosphoenrichment. For 3 biopsies, input to phosphoenrichment was slightly reduced but was accounted for by Progenesis/ SafeQuant (see below). These 3 samples did not show an outlier pattern in terms of the quantified phosphorylation sites after data processing and were included in subsequent analyses. The µRPLC–MS system was set up as described previously[92]. Chromatographic separation of peptides was carried out using an EASY nano-LC 1000 system (Thermo Fisher Scientific), equipped with a heated RP-HPLC column (75 µm × 37 cm) packed in-house with 1.9 µm C18 resin (Reprosil-AQ Pur, Dr. Maisch). Dried phosphopeptides were resuspended in 20 µl of 0.1% formic acid and 3 µl of the sample were injected per triplicate LC–MS/MS run. Samples were analyzed using a linear gradient ranging from 95% solvent A (0.15% formic acid, 2% acetonitrile) and 5% solvent B (98% acetonitrile, 2% water, 0.15% formic acid) to 30% solvent B over 90 min at a flow rate of 200 nl/min. Mass spectrometry analysis was performed on a Q-Exactive HF mass spectrometer equipped with a nanoelectrospray ion source (both Thermo Fisher Scientific). Each MS1 scan was followed by high-collision-dissociation (HCD) of the 10 most abundant precursor ions with dynamic exclusion for 20 s. Total cycle time was approximately 1 s. For MS1, 3e6 ions were accumulated in the Orbitrap cell over a maximum time of 100 ms and scanned at a resolution of 120,000 FWHM (at 200 m/z). MS2 scans were acquired at a target setting of 1e5 ions, accumulation time of 50 ms and a resolution of 30,000 FWHM (at 200 m/z). Singly charged ions and ions with unassigned charge state were excluded from triggering MS2 events. The normalized collision energy was set to 27%, the mass isolation window was set to 1.4 m/z and one microscan was acquired for each spectrum. The samples were measured in triplicates. The acquired raw-files were imported into the Progenesis QI software (v2.0, Nonlinear Dynamics Limited), which was used to extract peptide precursor ion intensities across all samples applying the default parameters. The generated mgf-files were searched using MASCOT (version 2.4.1) against a decoy database containing normal and reverse sequences of the predicted SwissProt entries of Homo sapiens (www.ebi.ac.uk) and commonly observed contaminants generated

using the SequenceReverser tool from the MaxQuant software (version 1.0.13.13). The search criteria were set as follows: full tryptic specificity was required; 3 missed cleavages were allowed; carbamidomethylation (C) was set as fixed modification; oxidation (M) and phosphorylation (STY) were applied as variable modifications; mass tolerance of 10 ppm (precursor) and 0.02 Da (fragments). The database search results were filtered using the ion score to set the false discovery rate (FDR) to 1% on the peptide and protein level, respectively, based on the number of reverse protein sequence hits in the datasets. The relative quantitative data obtained were normalized and statistically analyzed using SafeQuant[92]. Here the gMin algorithm was chosen. Afterwards, data were imported into FileMakerPro Advanced (Version FMP18) for further data processing. Imputed values were excluded and data were median subtracted per biopsy.

**Processing of phospho-proteome data.** SafeQuant[92] generated phospho-peptide centric quantifications. In order to generate quantitative data for single phosphorylation sites, peptides with more than one phosphorylation site were deconvoluted. As a next step all intensities assigned to a single phosphorylation site were added up to generate one cumulative intensity per phosphorylation site. The raw intensities of the triplicates were averaged and the mean values transformed by the logarithm to the base 2. Next, the values were normalized by median subtraction. The phospho-enrichments were performed and measured in three batches due to the limitation of the number of MS runs that can be performed using the same capillary column. In each batch also an aliquot of the pooled healthy sample was enriched and measured. Normalization to the pooled healthy sample was then performed batch-wise to the pooled healthy sample enriched and measured at the same time with the same batch. Localization probabilities of each phosphorylation site were determined per batch using ScaffoldPTM (Version 3.2.0) (Proteome Software) and the maximum observed localization probability observed was assigned to each phosphorylation site. Only phosphorylation sites with a minimum localization probability of 50% were taken into account.

We obtained data for 12205 phosphorylation sites (in 4230 proteins) that were quantified (always against healthy liver) in at least one HCC (Supplementary Fig. 2 and Supplementary Data 1), 9606 (3816) quantified in at least one HCC with >99% localization probability, 7911 (3160) quantified in at least 26 HCCs, 6403 (2856) quantified in at least 26 HCCs with >99% localization probability, 4112 (2031) quantified in all 51 HCCs and 3439 (1837) quantified in at all 51 HCCs with >99% localization probability. Starting from the 12205 phosphorylation sites, we removed proteins for which data were missing from >50% of samples (50% to enable sufficient data for imputation), resulting in 7893 phosphorylation sites from 3156 proteins for further analysis. Since data were generated over three batches, we corrected for the batch effect using the 'removeBatchEffect' function in the *edgeR* R package (v3.32.0)[93]. Data imputation using nearest neighbor averaging was performed using the 'impute.knn' function from the *impute* R package (v1.64.0). To normalize for overall protein levels, we computed the difference between the log2-fold-changes of phosphorylation site levels between tumors and the normal livers and the log2-fold-changes of protein levels between tumors and the normal livers, for proteins detected by both technologies.

**Differential expression analysis.** For transcriptome data, differential expression analysis was performed using the 'edgeR' package (v3.32.0)[93] between samples from a given class vs all other samples using raw RSEM expected counts as input. Specifically, normalization was performed using the "TMM" (weighted trimmed mean) method[94] and differential expression was assessed using the quasi-likelihood F-test, adjusted for multiple testing using Benjamini and Hochberg's method. For proteome and phosphoproteome data, differential expression analysis was performed using the 'limma' package (v3.46.0)[94], using the log2-fold-changes of protein/phosphorylation site levels between tumors and the normal livers. limma fits a linear model to compute the moderated t-statistics using a Bayesian model and adjusts the p-values for multiple testing using Benjamini and Hochberg's method. Genes, proteins and phosphorylation sites with adjusted $p \leq 0.05$ were considered differentially expressed.

**Pathway analysis.** Pathway analysis (over-representation analysis and gene set enrichment analysis (GSEA)) was performed using the 'clusterProfiler' (v3.18.0) and 'ReactomePA' (v1.34.0) packages[95,96] for KEGG/Reactome pathways and Gene Ontology biological processes subset. For proteome and phosphoproteome data, the corresponding sets of detected proteins were used as background for over-representation tests. $p \leq 0.05$ was considered statistically significant. Pathway analysis results were represented as barplots, dotplots or enrichment maps.

**Kinase-substrate enrichment analysis (KSEA).** KSEA[35] was performed using the 'KSEAapp' R package (v0.99.0)[97] using NetworKIN.cutoff = 5, the log fold-change and p-values computed from differential expression analysis (see "Differential expression analysis") of unimputed phosphorylation site levels and unimputed phosphorylation site levels normalized by overall protein levels (see "Processing of phospho-proteome data") as input.

**Analysis of dysregulated genes/proteins and pathways in HCC.** For the assessment of dysregulated genes and proteins in HCC, we performed differential expression analysis between HCCs and normal livers (see "Differential expression analysis") for transcriptome and proteome data. To compare the dysregulated genes/proteins between transcriptome data and proteome data, Uniprot accessions were mapped to Ensembl gene IDs, resulting in 5490 comparable genes/proteins. Dysregulated pathways were identified using a quadrant analysis, by performing over-representation tests (see "Pathway analysis").

**CNA-mRNA-protein correlation.** Correlation was performed using segmented log ratio for CNA, and the log2-fold-changes of protein levels between tumors and the median of normal livers for mRNA and protein data. Uniprot accessions were mapped to Ensembl gene IDs. For the CNA-mRNA correlation, 15272 genes were included. For the mRNA-protein correlation, 5481 genes were included. CNA-mRNA and mRNA-protein correlations were assessed using Spearman correlation tests. To assess the enrichment of genes within significant focal CNAs defined by GISTIC, genes were ranked according to Spearman correlation coefficient for GSEA analysis using the 'clusterProfiler' package[95]. p value ≤ 0.05 was considered statistically significant.

**Analysis of dysregulated phosphorylation sites in HCC.** For the assessment of dysregulated phosphorylation sites in HCC, we performed differential expression analysis between HCCs and normal livers (see "Differential expression analysis") for phosphorylation site levels with and without normalization by protein level. Significantly regulated phosphorylation sites (adjusted $p \leq 0.05$ and $|logFC| \geq 2$) were used for pathway analysis using over-representation tests (see "Pathway analysis"), separately for up- and downregulated phosphorylation sites, as well as for up- and down-regulated phosphorylation sites together. KSEA was also performed to identify differential kinase activity by computing the differential expression between HCCs and normal livers (see "Kinase-substrate enrichment analysis").

**Phosphoproteomic analysis for *CTNNB1* and *TP53* mutations.** Transcriptomic, proteomic and phosphoproteomic signatures of *CTNNB1* and *TP53* mutations were identified by differential expression analysis of the HCCs (see "Differential expression analysis"), by fitting a single model incorporating the mutation status of both genes. To identify phosphorylation sites associated with mutations in these two genes but were not associated with differences on the protein levels, we identified phosphorylation sites whose p-values were within the most extreme 10th percentile while the p-values of the corresponding proteins were within interquartile range. These phosphorylation sites were then subjected to pathway analysis by over-representation tests (see "Pathway analysis"). KSEA was also performed to identify differential kinase activity by computing the differential expression between mutant and wild-type HCCs (see "Kinase-substrate enrichment analysis").

**Single-omics clustering.** Identification of tumor subclasses based on somatic non-synonymous mutations was performed using oncosign (v1.0)[98] and Network-Based Stratification (pyNBS, downloaded on 4 June 2020)[99,100]. Significantly mutated genes identified using MutsigCV[36], as well as genes identified as significantly mutated in HCC in at least 2 of Martincorena et al.[38], Schultz et al.[40], Fujimoto et al.[3], Bailey et al.[39] (excluding *TERT*) and mutated (non-synonymous) in at least 3 tumor samples were included for the clustering.

Identification of tumor subclasses based on CNAs was performed using consensus k-means clustering and consensus hierarchical clustering using the 'ConsensusClusterPlus' R package (v1.54.0)[101], using gene-level amplification, gain, neutral, loss and deletion status as input. 117 tumors were included, excluding the 5 for which copy number gain/loss status could not be determined (see "Whole-exome sequencing and data processing"). For both clustering methods, Euclidean distance was used as the distance metric, and up to 10 clusters were evaluated over 100 subsamples. Hierarchical clustering was performed using the "ward.D2" method.

Identification of tumor subclasses based on transcriptome, proteome and phosphoproteome subclasses was performed using consensus nonnegative matrix factorization (NMF) and consensus k-means clustering using the 'CancerSubtypes' (v1.16.0) and the 'ConsensusClusterPlus' (v1.54.0) R packages[101,102], respectively. Log2-fold-change between tumors and the median of normal livers were used as input. For transcriptome and phosphoproteome clustering, features with standard deviation ≥2 across the tumors were included for clustering, resulting in 1370 and 1024 features, respectively. For proteome clustering, features with standard deviation ≥1 across the tumors were included, resulting in 1083 features. For consensus NMF, up to 10 clusters were evaluated over 50 NMF runs. For consensus k-means clustering, 1-Spearman correlation coefficient was used as the distance metric, and up to 10 clusters were evaluated over 100 subsamples.

Robustness of the subclasses was assessed by downsampling to 70%, 80% or 90% of samples over 20 runs, reclustering the reduced dataset, and calculating the adjusted Rand index compared to the full dataset. Cluster quality was assessed by Silhouette widths (except for mutation subclasses). The final number of clusters was chosen on the basis of the Silhouette widths and adjusted Rand index of the full dataset and for the 20 iterations of the downsampled datasets.

For mutation subclasses, the enrichment of mutated genes was assessed using a chi-squared test across all clusters and using Fisher's exact tests comparing a given cluster to all other clusters. For CNA subclasses, the enrichment of copy number-altered genes was assessed using Mann-Whitney U tests, adjusted for multiple

testing using Benjamini and Hochberg's method. Genes with adjusted $p \leq 0.05$ were considered statistically significant. For transcriptome, proteome and phosphoproteome subclasses, over-expressed features were identified by differential expression analysis between all samples in a given class and all other samples (see "Differential expression analysis") followed by pathway analysis (see "Pathway analysis"). KSEA was also performed for the phosphoproteomics subclasses (see "Kinase-substrate enrichment analysis"). Figures were generated using the 'ComplexHeatmap' R package (v2.6.2)[103].

**Integrative clustering**. Integrative clustering was performed for the 51 HCCs for which data were available for all data types using the 'iClusterBayes' function, which fits a Bayesian latent variable model that generates an integrated cluster assignment based on joint inference across data types, implemented in the *iClusterPlus* R package (v1.26.0)[56] and the similarity network fusion (SNF) method[57], which constructs a fusion patient similarity network by integrating the patient similarity obtained from each of the genomic data types, as implemented in the 'SNFtool' (v2.3.0) and the 'CancerSubtypes' R packages[57,102]. As input data, significantly mutated genes identified using MutsigCV[36], as well as genes identified as significantly mutated in HCC in at least 2 of Martincorena et al.[38], Schultz et al.[40], Fujimoto et al.[3], Bailey et al.[39] (excluding *TERT*) and mutated (non-synonymous) in at least 2 samples were included as mutational data. CNA data were included as collapsed copy number regions, constructed using the 'CNregions' function in the iClusterPlus R package to reduce the segmented logR ratio, as recommended in the package vignette, resulting in 927 features for clustering. For transcriptome and phosphoproteome data, features with standard deviation ≥2 across the tumors were included for clustering, resulting in 1646 and 1024 features, respectively. For proteome data, features with standard deviation ≥1 across the tumors were included, resulting in 1083 features. Transcriptome, proteome and phosphoproteome data were z-score-transformed prior to clustering. For both clustering, up to 10 clusters were evaluated.

Robustness of the subclasses was assessed by downsampling to 70%, 80% or 90% of samples over 20 runs, reclustering the reduced dataset, and calculating the adjusted Rand index compared to the full dataset. For iClusterBayes, the final number of clusters was chosen on the basis of the Bayesian Information Criterion, the deviance ratio (interpreted as percent explained variation) and adjusted Rand index between the full dataset and the 20 iterations of the downsampled datasets. For SNF, cluster quality was assessed by Silhouette widths and the final number of clusters was chosen on the basis of the Silhouette widths and adjusted Rand index of the full dataset and for the 20 iterations of the downsampled datasets.

The identification of enriched features were performed as per single-omics clustering for the corresponding data type. Figures were generated using the *ComplexHeatmap* R package (v2.6.2)[103].

**Analysis of the Cancer Genome Atlas (TCGA) data**. The Pan-cancer TCGA[104] data were obtained from https://gdc.cancer.gov/about-data/publications/pancanatlas. Specifically, the files "EBPlusPlusAdjustPANCANatlas_IlluminaHiSeq_RNASeqV2.geneExp.tsv" (RNA-seq), "TCGA-RPPA-pancan-clean.txt" (RPPA), "merged_sample_quality_annotations.tsv" (sample quality annotations) were obtained. Samples with 'cancer.type' as 'LIHC' and 'Do_not_use' as 'False' were retained. Six patients (TCGA-BC-A10Q, TCGA-DD-A3A6, TCGA-FV-A3I0, TCGA-GJ-A6C0, TCGA-KR-A7K2, TCGA-UB-A7MA) were removed as they were marked as "AWG_excluded_because_of_pathology". Edmondson grades were obtained from our previous publication[105]. RNA-seq gene expression data was log-transformed prior to principal component analysis. RPPA data was split into proteome and phosphoproteome according to whether the antibodies measure total protein or specific phosphorylation site, respectively.

**Cell Lines and transfection**. HCC-derived cell lines (Hep3B (ATCC Number: HB-8064), Huh6 (JCRB Cell Bank Number: JCRB0401), SNU449 (ATCC Number: CRL-2234), SNU182 (ATCC Number: CRL-2235), SNU398 (ATCC Number: CRL-2233) and Huh7 (JCRB Cell Bank Number: JCRB0403)) and hepatoblastoma derived cell line HepG2 (ATCC Number: HB-8065) were maintained in a 5% CO2-humidified atmosphere at 37 °C and cultured in Dulbecco's modified Eagle's medium (DMEM) supplemented with 10% fetal bovine serum, 1% penicillin/streptomycin (Bio-Concept, Allschwil, Switzerland), and 1% minimal essential medium–nonessential amino acids (ThermoFisher Scientific, Basel, Switzerland). Cell lines were confirmed negative for mycoplasma infection using the polymerase chain reaction (PCR)-based Universal Mycoplasma Detection kit (American Type Culture Collection, Manassas, VA). For transient *GPAM* knockdown, log-phase HepG2 and Huh7 cells were seeded at approximately 60% confluence in 6-well plates and transfected with siRNA against human *GPAM* (Dharmacon, #L-009946-00-0005) or non-targeting control siRNA (Dharmacon, #D-001810-10-20) to a final concentration of 25 nM, according to the manufacturer's protocol. Cells were harvested at 24, 48 and 72 h post transfection for protein isolation.

**Protein extraction and western blot**. Proteins were extracted using a Co-IP buffer. Cell lysates were then treated with 1× reducing agent (NuPAGE Sample Reducing Agent, #NP0009, Invitrogen, Carlsbad, CA), 1× loading buffer (NuPAGE LDS Sample Buffer, #NP0007 Invitrogen, Carlsbad, CA), boiled and loaded into neutral pH, pre-cast, discontinuous SDS-PAGE mini-gel system (NuPAGE Bis-Tris Protein Gels, ThermoFisher, Basel, Switzerland). After transfer onto a nitrocellulose membranes using the Trans-Blot Turbo Transfer System (Bio-Rad, Hercules, CA), proteins were detected using GPAM (G2617; 1:500, Santa Cruz,) and B-actin (A5441; 1:5000, Sigma, St. Louis, MO). Blots were scanned using the Odyssey Infrared Imaging System (LI-COR Biosciences, Lincoln, NE) and band intensity was quantified using ImageJ software (version 1.52a, LOCI, University of Wisconsin). The ratio of proteins of interest/loading control in treated samples were normalized to their counterparts in control cells.

**Cell proliferation assay**. Cell proliferation was assessed using CellTiter-Glo Luminescent Cell Viability Assay (#G7570, Promega). Statistical analyses were performed by the two-tailed unpaired Student's *t*-test using Prism software v6.0 (GraphPad Software).

**Statistical analysis**. Principal component analysis (PCA) of transcriptome, proteome and phosphoproteome data was performed using 'prcomp' from the *stats* R library. For transcriptome data, the upper-quartile-normalized RSEM values were used as input. For proteome and phosphoproteome data, the log2-fold-changes of protein/phosphorylation site abundance between tumors and the median of normal livers were used. Intra-group variability from PCA was computed as the pairwise Euclidean distance between samples of the same Edmondson grade using all principal components. Distance to normal livers was computed as the Euclidean distance between a given HCC sample to the median of normal livers using all principal components.

Statistical analyses of the clinicopathological variables were performed in R version 4.0.3. Comparisons of ordinal variables (e.g. BCLC, Edmondson grade, number of tumors) were performed using Mann–Whitney *U* tests. Comparisons of categorical variables (e.g. immunophenotype, presence of metastasis) were performed using Fisher's exact tests or chi-squared tests. Comparisons of numerical variables (e.g. tumor mutational burden) were performed using Mann–Whitney *U* or Kruskal–Wallis tests. Correction for multiple testing was performed using the Benjamini–Hochberg method.

The association of overall survival and molecular subclasses was performed using Cox proportional-hazards model, including BCLC stage as a covariate. Overall survival was defined as the time interval between the diagnosis of HCC to death. Individuals who were lost-to-follow-up or had undergone liver transplantation were considered censored. For patients with >1 biopsy included in the study, only one biopsy was considered if all biopsies were of the same molecular subclass and patients excluded from overall survival analysis if the biopsies were of multiple molecular subclasses. All statistical tests were two-sided unless otherwise indicated, and $p \leq 0.05$ was considered statistically significant.

**Reporting summary**. Further information on research design is available in the Nature Research Reporting Summary linked to this article.

## Data availability

Sequencing data generated in this study (including RAW sequencing data) are available under restricted access at the European Genome-phenome Archive under accessions EGAS00001005073 (whole-exome sequencing) and EGAS00001005074 (RNA-sequencing). Access is restricted because genetic data is personally identifiable. To obtain access and conditions of access to the EGA datasets, contact the corresponding author, who will respond within 4 weeks. The use of the data will be subjected to agreement of a data use policy, which details the minimum protection measures required related to data encryption and user access. The data will be available to the authorized users for the duration of the requested project. Users will have to specifically agree to preserve, at all times, the confidentiality of information and Data Subjects and to use or attempt to use the Data to compromise or otherwise infringe the confidentiality of information on Data Subjects and their right to privacy. User have to agree not to attempt to identify Data Subjects. The full data use policy will be available upon data access request. Hotspot mutations were obtained from http://www.cancerhotspots.org. Protein-coding genes for gene-level copy number analysis were obtained from Ensembl (http://grch37.ensembl.org/Homo_sapiens/Info/Index, release GRCh37.p13). Proteome and phosphoproteome data generated in this study are available in PRIDE (PXD025705 and PXD025836, respectively). For SWATH data analysis, the Uniprot SwissProt database (www.uniprot.org, 17th August 2015) was used for the searches. The publicly available Pan-cancer TCGA[104] data were obtained from https://gdc.cancer.gov/about-data/publications/pancanatlas. Specifically, the files "EBPlusPlusAdjustPANCAN_IlluminaHiSeq_RNASeqV2.geneExp.tsv" (RNA-seq), "TCGA-RPPA-pancan-clean.txt" (RPPA), "merged_sample_quality_annotations.tsv" (sample quality annotations) were obtained. Source data are provided with this paper. The remaining data are available within the Article, Supplementary Information or Source Data file. Source data are provided with this paper.

## Code availability

SafeQuant is available at GitHub [https://github.com/eahrne/SafeQuant].

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

## Acknowledgements

The project was funded by the ERC Synergy Grant 609883 (M.N.H. and M.H.H.). C.K.Y.N. and S.P. were supported by the Swiss Cancer Research foundation (KFS-4543-08-2018, KFS-4988-02-2020-R, respectively); L.M.T. was supported by AIRC (Fondazione AIRC per la Ricerca sul Cancro) grant number IG 2019 Id.23615, S.P. is supported by the Prof. Max Clöetta Foundation and the Surgery Department of the University Hospital Basel. The funders had no role in study design, data collection, and analysis, decision to publish, or preparation of the manuscript. Library preparations and sequencing were performed in the Genomics Facility Basel of the University of Basel and ETH Zurich Department of Biosystems Science and Engineering. Data analysis was performed at sciCORE scientific computing center at the University of Basel.

## Author contributions

L.M.T., M.N.H. and M.H.H. conceived and supervised the study; C.K.Y.N. performed the bioinformatic analysis; T.Bol. and M.H.H. performed the biopsy procedures; T.Bol., S.W. and M.H.H. provided the clinical information; C.E., M.S.M. and L.M.T. performed the histopathological assessment of tissue samples; X.W. and S.K. collected, processed and archived biopsy tissue; S.N., A.Su., M.A.M. and S.W. processed the tissue samples and performed nucleic acid extraction for sequencing; E.D., M.C. and M.N.H. performed the proteome and phosphoproteome profiling and data processing; T.Boc. and A.Sch. helped with the phosphoproteome profiling and data processing; C.K.Y.N., E.D. and M.C.-L. wrote the manuscript, which was initially revised by S.P. and M.H.H. All authors have read and revised the manuscript.

## Competing interests

The authors declare no competing interests.
