## [Peer review file · Nature Communications]

REVIEWER COMMENTS

Reviewer #1 (Remarks to the Author): Expert in proteomics and hepatocellular carcinoma

In the here presented manuscript, Charlotte et al. performed molecular characterization of a hepatocellular carcinoma (HCC) cohort (114 patients) by implementing integrative genomic, transcriptomic, proteomic and phosphoproteomic analyses. They identified deregulated pathways using differential expression analyses, suggesting the consistency and difference between mRNA and protein regulation. Phosphoproteomic analyses revealed that biological processes such as cell proliferation and DNA repair were deregulated on the phosphorylated modification level. Furthermore, they identified the candidate driver gene and evaluated the molecular changes associated with CTNNB1 and TP53 mutations. Subtypes of HCC determined by single-omics or an integrative analysis were associated with biological and clinical features of HCC.

This manuscript could be further strengthened in response to the following comments.

Major comments :

1. Proteomics and phosphoproteomics analyses were performed by data-independent manner by selected window acquisition of theoretical mass (SWATH) and the data-dependent and label-free manner, respectively. Quality control of this HCC cohort and quality assessments for MS data (to indicate the robust and accurate platform) should be described in the supplementary figures.
2. Previous studies have been well-characterized based on the surgically resected HBV-associated or early-staged HCCs. The authors in this study selected more extensive HCC biopsies across etiologies and clinical stages. They showed that low-grade HCCs were more homogeneous than high-grade HCCs. So further detailed studies should be performed using the proteogeomic data to identify the different molecular characterization of low- and high-grade or early- and late-stage of HCC. This may be helpful to reveal the molecular characteristics of HCC progression.
3. The authors identified 29 genes as candidate copy number-driven cancer genes by evaluated the correlation between copy number alteration (CNA), mRNA expression and protein expression. This should be further confirmed by evaluated the frequency of copy number alteration, the expression of mRNA and protein between the tumor and normal tissues and so on.
4. The study found a new significantly mutated gene ---GPAM, which has not been well identified in previous HCC study. Further analysis or validation should be added to confirm that it is a tumor suppressor.
5. Line 270, the author concluded that the EMT phenotype was frequently seen in CTNNB1-mutant HCCs, this should be further confirmed by analysis such as the marker expression in epithelial or mesenchymal cells, the EMT scores which represented the EMT phenotype between CTNNB1 mutant and WT HCCs.

6. Single-omics and multi-omics subtyping suggested different molecular features in HCC of this cohort. Comparison of the subtyping results with previous reported studies should be performed. Furthermore, are there any specific treatment modalities that could be applied among different clusters?

Minor comments

1. Figure 1f, why the authors used two different statistical test methods to identify differentially expressed genes/proteins ?
2. Copy number amplifications and deletions analysis results should be shown in the supplementary figures.
3. Line 162, regulation by TP53 is not included in Figure3b.
4. The legend of supplementary Figure 3 is missing and confused.
5. Line 181, increased AURKA activity in HCCs is not consistent with the supplementary Figure 3b results.
6. The impact of these 7 SMGs on their cognate products should be assessed using the RNA and protein data.
7. Representation of the figures need to be improved: e.g. Figure 1b (1d).
8. Unified writing: p-value, adj. P.val, adjusted.p and so on.
9. The project accession number of proteome and phosphoproteome data should be added in the section of Data Availability.
10. The description of Project PXD025836 in Proteomexchange shows that the authors analyzed the phospho-proteome of tumor and matched non-tumor biopsies from 51 treatment-naive Hepatocellular carcinoma (HCC) patients by DIA (SWATH). It is confused because the DDA was used for phosphoproteome in the manuscript.

Reviewer #2 (Remarks to the Author): Expert in hepatocellular carcinoma genomics

Ng et al. performed a proteogenomic analysis of HCC biopsies across etiologies and clinical stages. There are several issues:

1. The gap of knowledge and take-home message is unclear in this study.
 - a. As such, there is difficulty in appreciating the value/significance of the study in contributing to the field of HCC

b. Authors have only described that the study provides a comprehensive analysis of the proteomic and phosphoproteomic landscape of HCCs and that it identifies alterations within these two areas that could underlie HCC

c. Perhaps, further stating the how knowledge of such alterations may assist in future studies or addressing questions that have been previously raised may help bring light to the value of the study.

2. Besides the two articles related to HBV-HCC mentioned in the manuscript, studies which performed similar analysis in HCC were not discussed in this manuscript. For example, Zhang et al. also examined proteogenomic and copy number alteration in HCC (Zhang, Lou et al. 2019).

3. The background liver of HCC is usually complicated with hepatitis B virus (HBV), hepatitis C virus (HCV), alcoholic or non-alcoholic fatty liver diseases. Since the patient cohort is primarily alcohol liver disease (59%) and/or hepatitis C viral infection, the TCGA cohort could be used for validation of genomic, transcriptomic, and proteomic analysis of this study.

4. Authors may have made an error in the captions for Supplementary Figure 3 where Supplementary Figure 3c should be for 3d instead. Figure legend for Supplementary Figure 3c is missing.

a. Authors highlighted that ‘When analyzing dysregulated phosphorylation normalized by protein level, KSEA revealed increased AURKA, CDK1/2/5 and ERK1/2 but also GSK3B activity in HCCs (Supplementary Figure 3 and Supplementary Table 7).’, but based on the Supplementary Figure 3, AURKA appears to have a decreased activity instead of increased.

5. Authors described that they identified seven of the nine GPAM mutations were frameshift mutations, which strong suggests a tumor suppressor role. However, the link between frameshift mutations and tumor suppressor role is not further established nor cited.

6. Authors highlighted the associations of CTNNB1-mutants and TP53-mutants with lower and higher grade of tumors respectively, but the cited figure (Supplementary Figure 4) describes only the significance of association without indicating the direction of association.

7. In page 10, it is mentioned that “The proteome clusters, but not the phosphoproteome clusters, are associated with Edmondson grade. Further, significant association was observed between mutation, transcriptome and proteome clusterings (i.e. those associated with Edmondson grade), but not with the phosphoproteome clusters (Supplementary Figure 5).”

a. The same finding (“The proteome clusters, but not the phosphoproteome clusters, are associated with Edmondson grade”) is repeated twice.

b. The association of proteome clusters and phosphoproteome clusters with Edmondson grade were not shown in the Supplementary Figure 5 as this figure only shows association of mutation, copy number cluster and transcriptome with Edmondson grade (Supplementary Figure 5f, g and h).

c. The result might be more meaningful if authors are able to demonstrate that high gene expression /protein expression/copy number of specific genes or specific clusters/mutations is associated with higher Edmondson grade (worse clinical outcome) or vice versa instead of providing only the P value.

8. Authors demonstrated that the EMT phenotype frequently seen in CTNNB1-mutant HCCs may result from the altered phosphorylation of proteins involved in organization of actin filaments, which regulate

cell polarity and migration (Figure 4c). However, CTNNB1-mutant HCCs were not shown to be significantly associated with metastasis (Supplementary Figure 4). What is the possible reason for this difference?

9. Figure 4e was used to support the statement that 'Stathmin 2 (STMN2) and Nuclear receptor corepressor 1 (NCOR1) were overexpressed at the protein level, they were underexpressed on the mRNA level'. However, the under expression of STMN2 is not clear in Figure 4e.

10. Future work and limitation of this study should be discussed in the discussion.

Reference:

Zhang, Q., Y. Lou, J. Yang, J. Wang, J. Feng, Y. Zhao, L. Wang, X. Huang, Q. Fu, M. Ye, X. Zhang, Y. Chen, C. Ma, H. Ge, J. Wang, J. Wu, T. Wei, Q. Chen, J. Wu, C. Yu, Y. Xiao, X. Feng, G. Guo, T. Liang and X. Bai (2019). "Integrated multiomic analysis reveals comprehensive tumour heterogeneity and novel immunophenotypic classification in hepatocellular carcinomas." *Gut* 68(11): 2019-2031.

Reviewer #3 (Remarks to the Author): Expert in hepatocellular carcinoma

In the present study, Ng et al. conducted a proteogenomic analysis of human hepatocellular carcinoma (HCC) specimens with distinct clinical stages and etiologies. The authors found that pathways related to cell cycle, transcriptional and translational control, signal transduction, and metabolism were differentially regulated on genomic, transcriptomic, proteomic, and phosphoproteomic levels. In addition, a series of potential driver genes involved in the epithelial-to-mesenchymal transition, Wnt- β -catenin pathway, transcriptional control, and metabolism were identified. Of note, CTNNB1 mutations were found to be related to altered protein phosphorylation in actin filament components, whereas p53 mutations were associated with lipid metabolism. Thus, the authors conclude that HCC is a heterogeneous entity characterized by various tumor subgroups with distinct molecular and metabolic features.

The study by Ng et al. is novel, engaging, well-written, and provides critical insights into the genomic, proteomic, and metabolic features of human HCC. The study was adequately planned and conducted; state-of-the-art technologies were applied and related to each other, thus providing an extensive overview of the features of HCC subsets. The data are solid and fully support the conclusions drawn.

Minor issues:

- At least some of the findings should be confirmed in a representative number of cases via real-time RT-PCR, Western blotting, or immunohistochemistry.

- Silencing or overexpression experiments should be conducted in vitro to demonstrate the biological activity of some of the potential tumor driver genes identified.

REVIEWER COMMENTS

Reviewer #1 (Remarks to the Author): Expert in proteomics and hepatocellular carcinoma

In the here presented manuscript, Charlotte et al. performed molecular characterization of a hepatocellular carcinoma (HCC) cohort (114 patients) by implementing integrative genomic, transcriptomic, proteomic and phosphoproteomic analyses. They identified deregulated pathways using differential expression analyses, suggesting the consistency and difference between mRNA and protein regulation. Phosphoproteomic analyses revealed that biological processes such as cell proliferation and DNA repair were deregulated on the phosphorylated modification level. Furthermore, they identified the candidate driver gene and evaluated the molecular changes associated with CTNNB1 and TP53 mutations. Subtypes of HCC determined by single-omics or an integrative analysis were associated with biological and clinical features of HCC.

This manuscript could be further strengthened in response to the following comments.

Major comments :

1. Proteomics and phosphoproteomics analyses were performed by data-independent manner by selected window acquisition of theoretical mass (SWATH) and the data-dependent and label-free manner, respectively. Quality control of this HCC cohort and quality assessments for MS data (to indicate the robust and accurate platform) should be described in the supplementary figures.

Reply: We thank the reviewer for this comment. We have now included 2 new supplementary figures (**Supplementary Figures 2 and 3**) in the revised manuscript. Both the proteome and the phospho-proteome platforms show consistent and robust LC and MS performance, as shown in the total ion count (TIC) overview of the chromatograms and the number of quantified proteins or p-sites (**Supplementary Figure 2a and c**). Furthermore, the coefficients of variation (CV) of both proteome and phospho-proteome measurements lie in the expected range (**Supplementary Figures 2e and f**). The CV furthermore shows that the biological variation in our experiments is higher compared to the technical variation. **Supplementary Figure 2** therefore highlights the quantitative robustness and accuracy of our platforms.

As can be seen for the proteome by the spiked iRT peptides in **Supplementary Figure 3a-d**, we were able to detect on average eight data points per peak and the expected retention time was on average only 2-3 min delayed. The average mass shift variations were low (2-3 ppm) and a considerable number of the precursors/peptides/proteins in the spectral library could be quantified (22/23/59%). As can be seen for the phospho-proteome in **Supplementary Figure 3e-i**, we were able to acquire our data with an average LC peak width between 0-20 sec and could detect peptides in every retention time quartile of the chromatogram. Log₁₀ abundance of the detected peptides was on average between 4 and 5 and the m/z ratio of the detected peptides was on average between 400 and 600. Finally, we acquired our phospho-proteome data at an average mass shift of 2 ppm. These QC metrics highlight the technical robustness and accuracy of our proteomics and phospho-proteomics platforms.

2. Previous studies have been well-characterized based on the surgically resected HBV-associated or early-staged HCCs. The authors in this study selected more extensive HCC biopsies across etiologies and clinical stages. They showed that low-grade HCCs were more homogeneous than high-grade HCCs. So further detailed studies should be performed using the proteogeomic data to identify the different molecular characterization of low- and high-grade or early- and late-stage of HCC. This may be helpful to reveal the molecular characteristics of HCC progression.

Reply: We thank the reviewer for this suggestion. We compared the proteogenomic profiles of BCLC high (B-D) vs low (0 and A) HCCs. We observed no difference in terms of the frequency of mutated genes (**Supplementary Figure 7a**). On the transcriptomic and proteomic levels, high-stage HCCs overexpress genes and proteins related to cell cycle and mitosis, DNA repair and replication, transcriptional regulation. On the other hand, high-stage HCCs underexpress genes related to ECM formation and organization on the mRNA level and metabolism of fatty acids on the protein level (**Supplementary Figure 7b-c**). The results have also been included in **Supplementary Table 3**. We have also added the following text:

Results (page 4 paragraph 3)

Previous proteogenomic studies focused on surgically resected HBV-associated or early-stage HCCs^{2,11,12}, we therefore asked whether early and late-stage HCC would show different molecular characteristics. We observed no difference in terms of the frequency of mutated genes (**Supplementary Figure 7**). On the transcriptomic and proteomic levels, high-stage HCCs overexpress genes and proteins related to cell cycle and mitosis, DNA repair and replication, transcriptional regulation (**Supplementary Table 3**). On the other hand, high-stage HCCs underexpress genes related to ECM formation and organization on the mRNA level and metabolism of fatty acids on the protein level.

3. The authors identified 29 genes as candidate copy number-driven cancer genes by evaluated the correlation between copy number alteration (CNA), mRNA expression and protein expression. This should be further confirmed by evaluated the frequency of copy number alteration, the expression of mRNA and protein between the tumor and normal tissues and so on.

Reply: We thank the reviewer for this comment. The 29 genes were found in 5 GISTIC significant regions, each of which was gained/lost in between 27% and 61% of the cases. Of the 29 genes, 19 were differentially expressed on the mRNA level in the respective orientation (i.e. upregulated if amplified, downregulated if deleted, FDR<0.05), 11 were differentially expressed on the protein level, and 9 were differentially expressed on both levels (*ATP6V1C1*, *BYSL*, *CHD1L*, *NUDCD1*, *LRRC47*, *RRM2B*, *UBQLN4*, *XPO5*, *YWHAZ*). We have added these data to **Figure 2D** and **Supplementary Table 5**. We have revised the text to focus on this set of genes.

Abstract (page 2 paragraph 1)

We describe candidate copy number-driven driver genes involved in epithelial-to-mesenchymal transition, the Wnt- β -catenin, AKT/mTOR and Notch pathways, cell cycle and DNA damage regulation.

Results (page 5 paragraph 2)

To identify such candidate driver genes, we focused on the 136 genes that showed high CNA-mRNA and mRNA-protein correlation (Spearman $\rho > 0.5$) and specifically on the 29 within the 5 enriched GISTIC2 regions, which were gained or lost in 27%-61% of the cohort (**Figure 2d inset and Supplementary Tables 4 and 5**). We further narrowed down this list of 29 genes to those that were dysregulated in the expected orientation (i.e. upregulated in amplified regions and downregulated in deleted regions with respect to normal tissues, FDR<0.05). Of these 29, 19 were dysregulated on the mRNA level, 11 were dysregulated on the protein level, and 9 were dysregulated on both levels (*ATP6V1C1*, *BYSL*, *CHD1L*, *NUDCD1*, *LRRC47*, *RRM2B*, *UBQLN4*, *XPO5*, *YWHAZ*)... *BYSL* (Bystin Like, 6p21.1), a direct cMYC target²⁵, is required for nucleogenesis in HCC cell proliferation and its inhibition induces apoptosis, partially arrests the cell cycle and inhibits tumor formation in mouse xenografts²⁶. *BYSL* also reportedly promotes glioma/glioblastoma growth via the GSK-3 β / β -catenin and AKT/mTOR pathways^{27,28}.

Of note, there are genes with poorly characterized or controversial roles in HCC tumorigenesis among those that showed strong CNA-mRNA and mRNA-protein correlation. *ATP6V1C1* (ATPase H⁺ Transporting V1 Subunit C1, 8q22.3) is a member of the vacuolar ATPases (V-ATPases) family of proton pumps with roles in Wnt/ β -catenin, Notch, and mTOR signaling, as well as in the regulation of cell invasion, migration and metastasis²⁹. The subunit V1C (encoded by *ATP6V1C1*) is, in particular, upregulated in oral cancer, and its silencing impairs breast cancer growth and metastasis³⁰ but its role in HCC carcinogenesis is unclear. *RRM2B* (Ribonucleotide Reductase Regulatory *TP53* Inducible Subunit M2B, or p53R2, 8q22.3) is a p53 target and a regulator of DNA damage and replication stress³¹. *RRM2B* was reported to be downregulated in HCC and to inhibit cell migration and spreading through the Egr-1/PTEN/Akt1 pathway³². Yet its frequent amplification and overexpression in HCC and other cancers suggests it may promote oncogenesis^{33,34}, specifically in hypoxic conditions³⁵. The role of these genes/proteins in HCC may warrant further investigation.

Discussion (page 13 paragraph 1)

Among genes and proteins that show positive correlation on the CNA-mRNA and mRNA-protein levels and are dysregulated in HCC, we identified candidate driver genes, such as *NUDCD1*, *UBQLN4*, *BYSL*, *ATP6V1C1* and *RRM2B* involved in diverse processes including EMT, cell cycle and DNA damage regulation, and the regulation of the Wnt- β -catenin, AKT/mTOR and Notch pathways. At

least two of them have been suggested as therapeutic targets. Specifically, *UBQLN4* is reported to repress homologous recombination repair and promotes sensitivity to PARP1 inhibition²⁴, while *RRB2M*, a gene that appears to promote tumorigenesis in hypoxic conditions and encodes a component of the ribonucleotide reductase, may be targeted by gemcitabine³⁵, a chemotherapeutic agent sometimes used to treat HCC.

4. The study found a new significantly mutated gene ---GPAM, which has not been well identified in previous HCC study. Further analysis or validation should be added to confirm that it is a tumor suppressor.

Reply: We thank the reviewer for this comment. We performed knockdown experiments using siRNA in two HCC cell lines – HepG2 and Huh7. In both models, *GPAM* knockdown resulted in significantly increased cell proliferation, consistent with our conclusion that *GPAM* is likely a tumor suppressor. We have added these results to **Supplementary Figure 10**. We have amended the text as follows:

Results (page 8 paragraph 1)

Here we found seven of the nine *GPAM* mutations were frameshift mutations, strongly suggestive of a tumor suppressor role (**Supplementary Figure 10**). This is corroborated by experiments in the HepG2 and Huh7 HCC cell lines that knocking down *GPAM* significantly increased cell proliferation (**Supplementary Figure 10**).

Discussion (page 13 paragraph 1)

We showed that *GPAM* is a novel frequently mutated gene in HCC.

5. Line 270, the author concluded that the EMT phenotype was frequently seen in *CTNNB1*-mutant HCCs, this should be further confirmed by analysis such as the marker expression in epithelial or mesenchymal cells, the EMT scores which represented the EMT phenotype between *CTNNB1* mutant and WT HCCs.

Reply: We thank the reviewer for this important comment. Compared to normal livers, we observed a loss of E-cadherin in both *CTNNB1*-mutant and *CTNNB1*-wildtype tumors but the extent of E-cadherin loss was more pronounced in *CTNNB1*-mutant tumors. *KRT19* (Keratin-19) was similarly lower in *CTNNB1*-mutant compared to *CTNNB1*-wildtype tumors. This suggests a loss of the epithelial phenotype. However, we did not observe a corresponding increase in mesenchymal markers such as *TWIST1*, *VIM* (Vimentin) and *ZEB1* (**Figure 1 for reviewers** below), therefore inconsistent with our previous claim of an EMT phenotype in *CTNNB1*-mutant HCCs. The finding of a loss of the epithelial phenotype would still be consistent with the involvement of actin filament organization among the *CTNNB1*-mutant HCCs (**Figure 4b-c and Supplementary Figure 12**).

Figure 1 for reviewers

Loss of epithelial phenotype but no gain of mesenchymal phenotype in CTNNB1-mutant HCCs. Boxplots showing the expression of the epithelial markers (CDH1/E-cadherin and KRT19/Keratin-19) and the mesenchymal markers (TWIST1/Twist, VIM/Vimentin, CDH2/N-cadherin, ZEB1, FN1/fibronectin) in CTNNB1-mutant and -wildtype HCCs. Statistical comparisons between MUT and WT were performed using Mann-Whitney U tests. *: P<0.05. ns: not significant. Statistical comparisons between MUT and normal livers and WT and normal livers were performed using one-sided Mann-Whitney U tests. Orange boxes indicate P<0.05.

We have included the epithelial marker expression in **Supplementary Figure 12**. We have revised the text as follows:

Results (page 9 paragraph 2)

Indeed, we observed reduced expression of the epithelial markers *CDH1* (E-cadherin) and *KRT19* (Keratin-19) in the *CTNNB1*-mutant HCCs, suggesting a loss of the epithelial phenotype (**Supplementary Figure 12**).

Results (page 10 paragraph 2)

Taken together, the proteogenomic analysis suggests that the loss of epithelial phenotype seen in *CTNNB1*-mutant HCCs may result from alterations in phosphorylation of proteins involved in the organization of actin filaments, thereby regulating cell polarity and migration...

Discussion (page 13 paragraph 2)

We identified altered phosphorylation sites in proteins, such as ACF7, MAP1B, PDLIM2 and Par3-alpha, that may underpin the loss of epithelial phenotype frequently seen in *CTNNB1*-mutant HCCs.

6. Single-omics and multi-omics subtyping suggested different molecular features in HCC of this cohort. Comparison of the subtyping results with previous reported studies should be performed. Furthermore, are there any specific treatment modalities that could be applied among different clusters?

Reply: We compared our subtyping results with the previously published molecular subtypes by Hoshida et al. Similar to the associations with Edmondson histological grade, the mutations, transcriptomic and proteomic subtypes and the integrative clusters correlated with the Hoshida molecular subtypes but not the CNA and phosphoproteomic subtypes. These results have been added to **Supplementary Figure 13k**. The following text has been added:

Results (page 11 paragraph 1)

Similarly, the mutation and transcriptome clusters but not the CNA subclasses were associated with the Hoshida molecular subtypes⁶ (**Supplementary Figure 13**).

Results (page 11 paragraph 2)

The proteome clusters, but not the phosphoproteome clusters, are associated with Edmondson grade and the Hoshida molecular subtypes⁶ (**Figure 5a-b and Supplementary Figure 13**).

Results (page 12 paragraph 1)

Furthermore, we also found that the integrative clusters, as well as the single-omics clustering of mutation, transcriptome and proteome to be associated with the previously published molecular subtypes defined by Hoshida *et al.* (**Supplementary Figure 13**).

In regards to potential therapeutic targets, integrative cluster 2 is enriched for *CTNNB1* mutants, suggesting Wnt signaling may be a potential therapeutic target for tumors in this cluster. On the other hand, integrative cluster 3 is characterized by elevated cell cycle signatures and CDK1/2/6 activity. CDKs, in particular CDK4/6, are well-established therapeutic targets. We have added these points to the discussion:

Discussion (page 13 paragraph 1)

Integrative clustering of HCCs reveal that cluster 2 was enriched for *CTNNB1* mutants, suggesting Wnt signaling may be a potential therapeutic target for tumors in this cluster. On the other hand, integrative cluster 3 is characterized by elevated cell cycle signatures and CDK1/2/6 activity. CDKs, in particular CDK4/6, are well-established therapeutic targets. These results also suggest potential molecularly driven putative drug targets in specific HCC subsets.

Minor comments

1. Figure 1f, why the authors used two different statistical test methods to identify differentially expressed genes/proteins ?

Reply: We chose the most appropriate statistical models for the analysis of the different modalities. In this case, RNA-seq was counts data hence we chose a statistical model that would explicitly account for read counts, while our proteomics data was normalized log ratios, for which a statistical model based on read counts would not be appropriate.

2. Copy number amplifications and deletions analysis results should be shown in the supplementary figures.

Reply: We have added the amplifications and deletions identified by GISTIC to **Supplementary Figure 8**.

3. Line 162, regulation by TP53 is not included in Figure 3b.

Reply: We thank the reviewer for pointing that out. Regulation by TP53 is one of the enriched pathways (**Supplementary Table 7**) but was omitted from **Figure 3b**. That has now been added to the figure.

4. The legend of supplementary Figure 3 is missing and confused.

Reply: We apologise for the confusion. The legend for panel c (now **Supplementary Figure 9**) was missing. We have corrected that.

5. Line 181, increased AURKA activity in HCCs is not consistent with the supplementary Figure 3b results.

Reply: We thank the reviewer for pointing out this error. Indeed the reviewer is right. We have corrected the text to reflect the results as shown in the figure (now **Supplementary Figure 9d**).

6. The impact of these 7 SMGs on their cognate products should be assessed using the RNA and protein data.

Reply: We thank you for the suggestion. This has now been added to **Supplementary Figure 11**. We found that *APOB* and *TP53* mutations were associated with lower expression on the mRNA or the protein levels, while *CTNNB1* mutations were associated with increased expression. We have added the following text:

Results (page 8 paragraph 2)

An analysis of the SMGs on their cognate mRNA and protein products revealed that *APOB* and *TP53* mutations were associated with lower expression on the mRNA or the protein levels, while *CTNNB1* mutations were associated with increased expression (**Supplementary Figure 11**).

7. Representation of the figures need to be improved: e.g. Figure 1b (1d).

Reply: We apologise for the mislabeling of **Figure 1b** – this has now been fixed.

8. Unified writing: p-value, adj.P.val, adjusted.p and so on.

Reply: We have unified the terminologies to 'pvalue' and 'p.adjust' throughout the supplementary tables.

9. The project accession number of proteome and phosphoproteome data should be added in the section of Data Availability.

Reply: We have added the missing information. Apologies for the omission.

10. The description of Project PXD025836 in Proteomexchange shows that the authors analyzed the phospho-proteome of tumor and matched non-tumor biopsies from 51 treatment-naive Hepatocellular carcinoma (HCC) patients by DIA (SWATH). It is confused because the DDA was used for phosphoproteome in the manuscript.

Reply: We thank the reviewer for pointing this out. We have corrected the metadata of the data submission.

Reviewer #2 (Remarks to the Author): Expert in hepatocellular carcinoma genomics

Ng et al. performed a proteogenomic analysis of HCC biopsies across etiologies and clinical stages. There are several issues:

1. The gap of knowledge and take-home message is unclear in this study.

a. As such, there is difficulty in appreciating the value/significance of the study in contributing to the field of HCC

b. Authors have only described that the study provides a comprehensive analysis of the proteomic and phosphoproteomic landscape of HCCs and that it identifies alterations within these two areas that could underlie HCC

c. Perhaps, further stating the how knowledge of such alterations may assist in future studies or addressing questions that have been previously raised may help bring light to the value of the study.

Reply: We thank the reviewer for this comment. As the reviewer is aware, there have been 3 MS-based proteome studies and 2 of which included the study of the phosphoproteome. All three studies have focused on early-stage HBV-associated diseases. Furthermore, the phosphoproteome component in both of those studies was small, with the focus primarily on the proteome.

Our study generated high-quality proteome and phosphoproteome data (**Supplementary Figures 2 and 3**) for a much wider spectrum of HCCs across all clinical stages and major etiologies, thus more representative of the molecular heterogeneity of HCC. Given that many studies had already been performed on the genome and transcriptome of HCC, while relatively little is known on the (phospho)proteome level of the full spectrum of HCC, our emphasis on the proteome and, in particular, the phosphoproteome is therefore by design. However, we have also made extensive use of the genomic and transcriptomic data to identify correlations between the various molecular layers to determine the cellular and molecular processes regulated at each level. Through the analysis of the CNA-mRNA-protein correlation and the study of the phosphoproteome, we were able to nominate a number of putative drivers and drug targets that may warrant further study in the future.

We have revised the text to further emphasise the points above as follows:

Abstract (page 2 paragraph 1)

We performed an integrated proteogenomic analysis of hepatocellular carcinomas (HCCs) across clinical stages and etiologies. Pathways related to cell cycle, transcriptional and translational control, signaling transduction, and metabolism are dysregulated and differentially regulated on the genomic, transcriptomic, proteomic and phosphoproteomic levels. We describe candidate copy number-driven driver genes involved in epithelial-to-mesenchymal transition, the Wnt- β -catenin, AKT/mTOR and Notch pathways, cell cycle and DNA damage regulation. The targetable aurora kinase A and CDKs

are upregulated. *CTNNB1* and *TP53* mutations are associated with altered protein phosphorylation related to actin filament organization and lipid metabolism, respectively. Integrative proteogenomic clusters show that HCC constitutes heterogeneous subgroups with distinct regulation of biological processes, metabolic reprogramming and kinase activation. Our study provides a comprehensive overview of the proteomic and phosphoproteomic landscapes of HCCs, revealing the major pathways altered in the (phospho)proteome.

Introduction (page 2 paragraph 2)

More recently, three mass spectrometry-based proteogenomic studies of hepatitis B virus (HBV)-associated HCCs have been published¹¹⁻¹³.... While these studies have provided great insights into the proteome of HCCs that are primarily early-stage and HBV-associated, relatively little of their phosphoproteome was described. Here we performed an integrated (phospho)proteogenomic analysis of HCC biopsies across etiologies and clinical stages, representative of the wide spectrum of molecular heterogeneity of HCC.

Discussion (page 15 paragraph 2)

Our study generated high-quality proteome and phosphoproteome data (**Supplementary Figures 2 and 3**) for a wide spectrum of HCCs across all clinical stages and major etiologies, thus more representative of the molecular heterogeneity of HCC than previous studies¹¹⁻¹³. In particular, our analysis of the phosphoproteome has provided much deeper insights into the major processes and kinases altered in HCC compared to previous studies. Our analyses of the CNA-mRNA-protein correlation, phosphoproteome and integrative clusters of HCCs have also nominated a number putative drivers and drug targets that may warrant further study in the future.

2. Besides the two articles related to HBV-HCC mentioned in the manuscript, studies which performed similar analysis in HCC were not discussed in this manuscript. For example, Zhang et al. also examined proteogenomic and copy number alteration in HCC (Zhang, Lou et al. 2019).

Reply: We thank the reviewer for pointing out the omission. We have added the following text:

Introduction (page 2 paragraph 2)

More recently, three proteogenomic studies of hepatitis B virus (HBV)-associated HCCs have been published¹¹⁻¹³.... In the third study, the authors found significant intratumor heterogeneity on the genome and transcriptome levels but patient specificity on the proteome and metabolome levels among resected and predominantly early-stage and HBV-associated HCCs¹³.

Results (page 5 paragraph 1)

We similarly observed a higher fraction of genes with significantly positive (Spearman $\rho > 0.3$) mRNA-protein correlation than for CNA-mRNA correlation (45.1% vs 32.0%), with the latter higher than previously reported¹³.

Discussion (page 14 paragraph 1)

Finally, as previously reported¹³, the presence of *TP53* mutations is associated with loss of expression on the mRNA level.

3. The background liver of HCC is usually complicated with hepatitis B virus (HBV), hepatitis C virus (HCV), alcoholic or non-alcoholic fatty liver diseases. Since the patient cohort is primarily alcohol liver disease (59%) and/or hepatitis C viral infection, the TCGA cohort could be used for validation of genomic, transcriptomic, and proteomic analysis of this study.

Reply: We thank the reviewer for the comment and suggestion. On the basis of the molecular data, we did not observe segregations according to the underlying liver diseases (new **Supplementary Figure 5**). As the reviewer suggested, we further analyzed the TCGA data, which covers a wider range of underlying liver diseases. Principal component analyses of the RNA-seq-based transcriptome data, as well as the RPPA-based proteome and phosphoproteome data, of the TCGA cohort support our conclusion that higher-grade HCCs are more heterogeneous than lower-grade HCCs (new **Supplementary Figure 6**). Given the limited number of proteins and phosphoproteins covered by the RPPA (<180) and the fact that only HCCs (no normal livers) were profiled on the RPPA, we were unable to use this data to validate many of our analyses. However, we note that some of our findings are in agreement with other HCC studies predominantly associated with HBV.

For example, our integrative subclasses resemble the proteome classes identified in HBV cohorts. Overall, we believe that our findings are applicable to HCCs of diverse etiologies.

Results (page 4 paragraph 1)

HCCs did not segregate according to their underlying liver diseases (**Supplementary Figure 5**).... These observations were corroborated by an analysis of The Cancer Genome Atlas data (**Supplementary Figure 6**).

4. Authors may have made an error in the captions for Supplementary Figure 3 where Supplementary Figure 3c should be for 3d instead. Figure legend for Supplementary Figure 3c is missing.

Reply: We thank the reviewer for noticing the error. Indeed the figure legend for panel c (now **Supplementary Figure 9**) was missing. We have corrected it.

a. Authors highlighted that ‘When analyzing dysregulated phosphorylation normalized by protein level, KSEA revealed increased AURKA, CDK1/2/5 and ERK1/2 but also GSK3B activity in HCCs (Supplementary Figure 3 and Supplementary Table 7).’, but based on the Supplementary Figure 3, AURKA appears to have a decreased activity instead of increased.

Reply: We thank the reviewer for pointing out this error. Indeed the reviewer is right. We have corrected the text to reflect the results as shown in the figure (now **Supplementary Figure 9d**).

5. Authors described that they identified seven of the nine GPAM mutations were frameshift mutations, which strongly suggests a tumor suppressor role. However, the link between frameshift mutations and tumor suppressor role is not further established nor cited.

Reply: We thank the reviewer for this comment. We performed knockdown experiments using siRNA in two HCC cell lines – HepG2 and Huh7. In both models, *GPAM* knockdown resulted in significantly increased cell proliferation, consistent with our conclusion that *GPAM* is likely a tumor suppressor. We have added these results to **Supplementary Figure 10** and the following sentences to the text.

Results (page 8 paragraph 1)

Here we found seven of the nine *GPAM* mutations were frameshift mutations, strongly suggestive of a tumor suppressor role (**Supplementary Figure 10**). This is corroborated by experiments in the HepG2 and Huh7 HCC cell lines that knocking down *GPAM* significantly increased cell proliferation (**Supplementary Figure 10**).

Discussion (page 13 paragraph 1)

We showed that *GPAM* is a novel frequently mutated gene in HCC.

6. Authors highlighted the associations of CTNNB1-mutants and TP53-mutants with lower and higher grade of tumors respectively, but the cited figure (Supplementary Figure 4) describes only the significance of association without indicating the direction of association.

Reply: Indeed. We have added a supplementary table to show the direction of association (**Supplementary Table 10**). We thank you for the suggestion.

7. In page 10, it is mentioned that “The proteome clusters, but not the phosphoproteome clusters, are associated with Edmondson grade. Further, significant association was observed between mutation, transcriptome and proteome clusterings (i.e. those associated with Edmondson grade), but not with the phosphoproteome clusters (Supplementary Figure 5).”

a. The same finding (“The proteome clusters, but not the phosphoproteome clusters, are associated with Edmondson grade”) is repeated twice.

Reply: We thank the reviewer for pointing this out. We took out the repeated statement.

b. The association of proteome clusters and phosphoproteome clusters with Edmondson grade were not shown in the Supplementary Figure 5 as this figure only shows association of mutation, copy number cluster and transcriptome with Edmondson grade (Supplementary Figure 5f, g and h).

Reply: We now show Edmondson grade in **Figures 5a and 5b**.

c. The result might be more meaningful if authors are able to demonstrate that high gene expression/protein expression/copy number of specific genes or specific clusters/mutations is associated with higher Edmondson grade (worse clinical outcome) or vice versa instead of providing only the P value.

Reply: We have modified **Figure 5 and Supplementary Figure 13f-h** to clearly show the association between the specific clusters and Edmondson grade and BCLC. We have also added the following sentences to the text:

Results (page 11 paragraph 1)

Of note, mutation cluster 3 and transcriptome cluster 1 were associated with high Edmondson grade.

Results (page 11 paragraph 2)

In particular, proteome cluster 1 was enriched for high-grade and high-stage HCCs.

Results (page 11 paragraph 3)

Integrative clusters 2 and 3 were associated with low- and high-grade HCCs, respectively.

8. Authors demonstrated that the EMT phenotype frequently seen in *CTNNB1*-mutant HCCs may result from the altered phosphorylation of proteins involved in organization of actin filaments, which regulate cell polarity and migration (Figure 4c). However, *CTNNB1*-mutant HCCs were not shown to be significantly associated with metastasis (Supplementary Figure 4). What is the possible reason for this difference?

Reply: We thank the reviewer for this question. Our data showed that *CTNNB1*-mutant HCCs are associated with the loss of epithelial phenotype (rather than EMT as previously stated). Nonetheless, it is also true that *CTNNB1*-mutant HCCs are not associated with metastasis. We believe there are two main reasons for this. First, only ~10% of the patients had metastasis, hence the number was relatively small. Second, The *CTNNB1*-wild-type group is heterogeneous, hence we may not observe differences when comparing *CTNNB1*-mutant HCCs to such a heterogeneous group.

9. Figure 4e was used to support the statement that 'Stathmin 2 (STMN2) and Nuclear receptor corepressor 1 (NCOR1) were overexpressed at the protein level, they were underexpressed on the mRNA level'. However, the under expression of STMN2 is not clear in Figure 4e.

Reply: We thank the reviewer for this comment. We have now shown the z-score-transformed data in **Figures 4a and 4e**, which should make it easier to see the differences.

10. Future work and limitation of this study should be discussed in the discussion.

Reply: We thank the reviewer for this comment. We have added the following passage to the Discussion.

Discussion (page 14 paragraph 3)

Proteomic and phosphoproteomic profiling technologies have improved significantly over the past decade but are not without challenges. In our study, we performed (phospho)proteomic profiling for about half of the samples but in triplicates to ensure robust statistics. Compared to WES and RNA-seq, (phospho)proteome data are relatively sparse and not all proteins and phosphorylation sites can be detected. It should also be noted that not all proteins are expressed at a given time at detectable levels and the detection of phosphorylation sites is challenging due to the high degree of variability between tissues and samples. Further, some phosphorylation sites may never be detected as they are on hard-to-detect peptides. Our study has also generated interesting leads for future studies. Mutagenesis experiments would be required to properly dissect the effect of *CTNNB1* and *TP53* mutations on the phosphoproteome. The mechanism of *GPAM* mutations would also require further experiments for full characterization.

Reviewer #3 (Remarks to the Author): Expert in hepatocellular carcinoma

In the present study, Ng et al. conducted a proteogenomic analysis of human hepatocellular carcinoma (HCC) specimens with distinct clinical stages and etiologies. The authors found that pathways related to cell cycle, transcriptional and translational control, signal transduction, and metabolism were differentially regulated on genomic, transcriptomic, proteomic, and phosphoproteomic levels. In addition, a series of potential driver genes involved in the epithelial-to-mesenchymal transition, Wnt- β -catenin pathway, transcriptional control, and metabolism were identified. Of note, *CTNNB1* mutations were found to be related to altered protein phosphorylation in actin filament components, whereas p53 mutations were associated with lipid metabolism. Thus, the authors conclude that HCC is a heterogeneous entity characterized by various tumor subgroups with distinct molecular and metabolic features.

The study by Ng et al. is novel, engaging, well-written, and provides critical insights into the genomic, proteomic, and metabolic features of human HCC. The study was adequately planned and conducted;

state-of-the-art technologies were applied and related to each other, thus providing an extensive overview of the features of HCC subsets. The data are solid and fully support the conclusions drawn.

Authors: We thank the reviewer for the positive feedback.

Minor issues:

- At least some of the findings should be confirmed in a representative number of cases via real-time RT-PCR, Western blotting, or immunohistochemistry.

Reply: We thank the reviewer for this comment. We believe that the RNA-seq technology itself and the associated bioinformatics methods are sufficiently mature these days that secondary techniques such as qPCR are no longer required to confirm the observations made on RNA-seq. Similarly for the proteome experiments, we do not think Western blotting or immunohistochemistry would add substantial information to the proteomic analysis. Unlike the unbiased approach of untargeted mass spectrometry, the outcomes of both Western blotting and immunohistochemistry highly depend on the performance of the specific antibodies and are much less sensitive. Moreover, immunohistochemistry is only a semi-quantitative technique.

- Silencing or overexpression experiments should be conducted in vitro to demonstrate the biological activity of some of the potential tumor driver genes identified.

Reply: We thank the reviewer for this comment. We performed knockdown experiments using siRNA in two HCC cell lines – HepG2 and Huh7. In both models, *GPAM* knockdown resulted in significantly increased cell proliferation, consistent with our conclusion that *GPAM* is likely a tumor suppressor. We have added these results to **Supplementary Figure 10**. We have also added the following to the text:

Results (page 8 paragraph 1)

Here we found seven of the nine *GPAM* mutations were frameshift mutations, strongly suggestive of a tumor suppressor role (**Supplementary Figure 10**). This is corroborated by experiments in the HepG2 and Huh7 HCC cell lines that knocking down *GPAM* significantly increased cell proliferation (**Supplementary Figure 10**).

REVIEWERS' COMMENTS

Reviewer #1 (Remarks to the Author):

The authors experimentally addressed the majority of the raised concerns and clarified several aspects raised during the revision of the manuscript.

Reviewer #2 (Remarks to the Author):

As there are other proteomic and phosphoproteomic studies on HCC already, the value of this study remains unclear.

The link between their frameshift mutations and tumor suppressor role is still unclear. To address this, only siRNA knockdown was performed instead of recapitulating the frameshift mutation

Reviewer #3 (Remarks to the Author):

The authors have sufficiently addressed the concerns raised by the Reviewer. Note that the HepG2 is a hepatoblastoma and not an HCC cell line (PubMed ID 19751877). Please modify the text accordingly.

We thank the reviewers for their insightful and helpful comments.

Point-to-point reply:

REVIEWERS' COMMENTS

Reviewer #1 (Remarks to the Author):

The authors experimentally addressed the majority of the raised concerns and clarified several aspects raised during the revision of the manuscript.

Reply: we thank the reviewer.

Reviewer #2 (Remarks to the Author):

As there are other proteomic and phosphoproteomic studies on HCC already, the value of this study remains unclear.

Reply: There are several aspects of our work that makes it different from previous proteomic and phosphoproteomic studies. All three published studies have focused on early-stage HBV-associated diseases. Our study generated high-quality proteome and phosphoproteome data for a much wider spectrum of HCCs across all clinical stages and major etiologies, thus more representative of the molecular heterogeneity of HCC. Given that many studies had already been performed on the genome and transcriptome of HCC, while relatively little is known on the (phospho)proteome level of the full spectrum of HCC, our emphasis on the proteome and, in particular, the phosphoproteome is therefore by design. However, we have also made extensive use of the genomic and transcriptomic data to identify correlations between the various molecular layers to determine the cellular and molecular processes regulated at each level. Through the analysis of the CNA-mRNA-protein correlation and the study of the phosphoproteome, we were able to nominate a number of putative drivers and drug targets that may warrant further study in the future. We have emphasised these points in our original revision of the manuscript.

The link between their frameshift mutations and tumor suppressor role is still unclear. To address this, only siRNA knockdown was performed instead of recapitulating the frameshift mutation

Reply: The siRNA-knockdown analysis that we provide in the revised manuscript clearly demonstrates the growth inhibitory function of GPAM. We think that it is reasonable to assume that most frameshift mutations have detrimental effects on protein structure and function. We therefore think that our siRNA-knockdown supports the proposed role of a tumor suppressor for GPAM. We agree with the reviewer that to demonstrate the effect of each of the 7 frameshift mutations in GPAM described in our manuscript, we would have to individually introduce them in the wildtype gene, express the mutated genes (on a GPAM null background), and functionally test them. However, we think that such a detailed

analysis of the functional consequences of the 7 frameshift mutations is beyond the scope of our manuscript.

Reviewer #3 (Remarks to the Author):

The authors have sufficiently addressed the concerns raised by the Reviewer. Note that the HepG2 is a hepatoblastoma and not an HCC cell line (PubMed ID 19751877). Please modify the text accordingly.

Reply: We thank the reviewer for his comment, and have corrected the manuscript accordingly.